# An open label, block randomized, community study of the safety and efficacy of co-administered ivermectin, diethylcarbamazine plus albendazole *vs*. diethylcarbamazine plus albendazole for lymphatic filariasis in India

**Purushothaman Jambulingam**[1], **Vijesh Sreedhar Kuttiatt**[1]*,
**Kaliannagounder Krishnamoorthy**[1], **Swaminathan Subramanian**[1]*,
**Adinarayanan Srividya**[1], **Hari Kishan K. Raju**[1], **Manju Rahi**[2], **Roopali K. Somani**[3],
**Mallanna K. Suryaprakash**[4], **Gangeshwar P. Dwivedi**[5], **Gary J. Weil**[6]

1 ICMR-Vector Control Research Centre, Puducherry, India, 2 Indian Council of Medical Research, New Delhi, India, 3 Department of Clinical Pharmacology & Therapeutics, Nizams Institute of Medical Sciences, Hyderabad, India, 4 District Vector-Borne Disease Control Programme, Yadgiri, Karnataka, India, 5 Consultant (Project), Yamuna Vihar, Himachal Pradesh, India, 6 Washington University School of Medicine, St. Louis, Missouri, United States of America

* vijeshvcrc.icmr@gmail.com (VSK); ssubra@yahoo.com, ssubra@1958@gmail.com (SS)

# Abstract

## Background

Better drug regimens for mass drug administration (MDA) could accelerate the Global Programme to Eliminate Lymphatic Filariasis (LF). This community study was designed to compare the safety and efficacy of MDA with IDA (ivermectin, diethylcarbamazine and albendazole) or DA (diethylcarbamazine and albendazole) in India.

## Methodology/Principal findings

This two-armed, open-labelled, block randomised, community study was conducted in LF endemic villages in Yadgir district, Karnataka, India. Consenting participants ≥5 years of age were tested for circulating filarial antigenemia (CFA) and microfilaremia (Mf) before treatment with a single oral dose of IDA or DA. Adverse events (AEs) were monitored actively for two days and passively for five more days. Persons with positive CFA or Mf tests at baseline were retested 12-months post-treatment to assess treatment efficacy.

Baseline CFA and Mf-rates were 26.4% and 6.9% in IDA and 24.5% and 6.4% in DA villages respectively. 4758 and 4160 participants received IDA and DA. Most AEs were mild after both treatments; fewer than 0.1% of participants experienced AEs with severity > grade 1. No serious AEs were observed. Fever, headache and dizziness were the most common AEs. AE rates were slightly higher after IDA than DA (8.3% vs. 6.4%, P<0.01). AEs were more frequent in females and Mf-positives after either treatment, but significantly more frequent after IDA (40.5% vs 20.2%, P < 0.001).

**Data Availability Statement:** The data from this study are available at Washington University in St. Louis, URL https://digitalcommons.wustl.edu/open_access_pubs/9999/.

**Funding:** This study was supported in part by grant OPPGH5342 from the Bill & Melinda Gates Foundation to Washington University. The study was also supported in part by the Coalition for Operational Research on Neglected Tropical Diseases Support Centre, which is funded at the Task Force for Global Health primarily by the Bill & Melinda Gates Foundation, by the United Kingdom Department for International Development, and by the United States Agency for International Development through its Neglected Tropical Diseases Program. Ivermectin was donated by Merck Sharp Dohme (MSD), also known as Merck & Co. (Kenilworth, NJ, USA). Albendazole (produced and donated by GlaxoSmithKline) and diethylcarbamazine (DEC, produced and donated by Eisai Co.) were obtained from the stocks of Ministry of Health and Family Welfare, Govt of India. The funders and drug donors had no role in study design, data collection and analysis, decision to publish, or preparation of the manuscript.

**Competing interests:** The authors have declared that no competing interests exist.

IDA was more effective for clearing Mf than DA (84% vs. 61.8%, P < 0.001). Geometric mean Mf counts per 60μl in retested Mf-positives decreased by 96.4% from 11.8 after IDA and by 90.0% from 9.5 after DA. Neither treatment was effective for clearing CFA.

## Conclusions/Significance

IDA had an acceptable safety profile and was more effective for clearing Mf than DA. With adequate compliance and medical support to manage AEs, IDA has the potential to accelerate LF elimination in India.

## Trial registration

Clinical Trial Registry of India (CTRI No/2016/10/007399)

## Author summary

Lymphatic filariasis (LF) is a major neglected tropical disease that is caused by filarial nematode worms. The strategies of the Global Programme to Eliminate Lymphatic Filariasis, launched in 2000, are mass drug administration (MDA) of antifilarial medications to kill the parasites and reduce transmission and morbidity management and disability prevention for those who are already affected by the disease. Recent clinical trials have shown that a single co-administered dose of ivermectin, diethylcarbamazine and albendazole (IDA) is more effective for clearing microfilariae (Mf) from the blood than the traditional two-drug regimen (DA). That is important, because blood Mf are essential for mosquitoes to transmit the parasite. As part of a large multicenter study, we assessed the safety of IDA and compared the efficacy of IDA and DA for clearing parasites from the blood. We treated almost 9,000 people in *Wuchereria bancrofti* endemic villages with either IDA or DA. Adverse events (AE) were monitored actively for two days and passively for another five days. AE rates were slightly higher after IDA than DA, but AEs were mild and self-limited. Infected persons, adults and females had higher AE rates in both treatment areas. We retested infected persons one year after treatment. IDA was significantly more effective for clearing Mf and reducing blood Mf counts than DA. Neither treatment was effective for clearing circulating filarial antigenemia. Our large study showed that IDA was well tolerated and more effective than DA. This new treatment has the potential to hasten LF elimination in India and many other countries.

## Introduction

Lymphatic filariasis (LF) is a major public health problem in many countries in the tropics and subtropics including India. LF, caused by three filarial parasite species (namely, *Wuchereria bancrofti*, *Brugia malayi* and *Brugia timori*), is endemic in 72 countries [1]. Bancroftian filariasis in India accounts for approximately one-third of the global LF infection burden. The Global Programme to Eliminate Lymphatic Filariasis (GPELF) was launched in 2000, with mass drug administration (MDA) as the recommended preventive chemotherapy strategy to interrupt transmission, and morbidity management to alleviate the suffering of chronic cases. In 2014, 72 countries were considered endemic and about 1.1 billion people required MDA [1]. India alone accounts for about 57% of those who require MDA [2]. Annual rounds of MDA with a

single dose of diethylcarbamazine (DEC) at 6mg/Kg body weight and albendazole with a flat dose of 400mg (DA) is the recommended strategy of intervention for most endemic countries outside of Africa, while ivermectin plus albendazole (IA) is the recommended regimen for countries where LF is co-endemic with onchocerciasis [3]. GPELF has made significant progress. It has delivered more than seven billion treatments to more than 910 million people, 14 countries have been validated to have achieved elimination of LF as a public health problem, and MDA has been stopped and post-MDA surveillance is in progress in another 10 countries [4]. The programme has reduced the number of countries that require MDA to 49 and reduced the population at risk for LF by 42%. As of 2018, about 892.9 million people were living in areas with ongoing transmission that still required MDA [4].

India launched its national LF elimination programme in 2004 as a pilot program that implemented MDA with an annual dose of DEC alone in 14 districts of 7 states. By 2008, the program had achieved 100% geographic coverage in the country by providing MDA with DEC plus albendazole (DA) in all 256 known endemic districts in 21 states and union territories. The programme has made significant progress since that time, and MDA has been halted in 99 districts that moved into post-MDA surveillance according to WHO guidelines [2]. However, the programme will not meet the deadline for achieving elimination goals before 2020, as Mf prevalence remains > 1% in some "hard-core" districts. There is a need for additional tools to accelerate interruption of transmission in these districts to achieve the goal of national LF elimination in the next few years.

A pilot study in Papua New Guinea (PNG) tested a novel single dose triple-drug regimen against filariasis that added ivermectin (200 μg per kg body weight) to the standard double drug regimen of diethylcarbamazine and albendazole (IDA) [5]. A single dose of IDA resulted in complete suppression of microfilaremia in all 12 study participants one year after treatment. However, IDA treatment resulted in an increased frequency of mild to moderate adverse events in that study compared to DEC plus albendazole alone (DA). A larger trial in PNG showed that a single dose of IDA cleared Mf for at least three years in 96% of 60 participants [6]. An added benefit of adding ivermectin is its broad-spectrum coverage against soil transmitted helminthic infections and its activity against ectoparasites such as lice and scabies mites [7,8].

A modelling study based on LF parameters for India suggested that areas in India with high baseline infection prevalence and low MDA compliance might require as many as 12 rounds of MDA with the standard DA regimen to achieve elimination [9]. Moreover, a recent simulation modelling study suggested that for areas in India with 5% residual Mf prevalence after 10 annual rounds of MDA with DA, IDA could lead to local elimination with three annual rounds, if compliance of at least 65% could be achieved [10,11].

Although results of small clinical trials had shown exciting results, IDA still required rigorous safety and efficacy testing in different settings before it could be recommended for large-scale use in LF elimination programmes. At the time of initiation of this study in 2016, no large-scale data were available on the safety and efficacy of IDA in community settings. WHO protocols for changing guidelines for public health programs require safety data for new treatments from at least 10,000 people across multiple settings [12]. In this context, the DOLF (Death to Onchocerciasis and Lymphatic Filariasis) project coordinated a multi-country study (India, Indonesia, Fiji, Haiti and Papua New Guinea) to collect data to support policy change at WHO. A combined analysis of the safety data from all five country sites with different endemic settings has been published recently [13], but that report did not include efficacy data.

We now report a detailed analysis of the safety and efficacy data collected from the IDA safety study that was performed in India. This was the largest of the community IDA safety

studies, and it was performed in an area with high endemicity despite many years of prior MDA. The purpose of the current study was to assess the safety and efficacy of IDA as an MDA regimen in comparison to that of DA, the reference regimen used by India's LF elimination programme, which is the largest in the world.

## Methods

### Ethics statement and regulatory monitoring

The study was approved by the Institutional Human Ethics Committee of the ICMR-Vector Control Research Centre (IHEC/IRB No.: IHEC-0316/RJ dated 13 April 2016), and study progress was periodically reported to the Committee. The study was registered with the Clinical Trial Registry of India (CTRI No/2016/10/007399). A written informed consent was obtained from all participants aged 18 years and older before any study procedures or drug administration was performed. Enrolment of minors (age 7–17 years of age) was done after obtaining written informed consent from at least one parent or legal guardian plus verbal assent from minors aged 7–17 year old. For participation of children aged 5–6 years, a formal written consent was obtained from one of the parents or legal guardians.

An independent medical monitor followed the safety data, and a Data Safety Monitoring Board (DSMB-India) periodically reviewed the safety data.

### Study location

The study was conducted in villages of Yadgir district in Karnataka state in South India (Fig 1). This semi-arid, economically challenged district is endemic for bancroftian filariasis transmitted by *Culex quinquefasciatus* mosquitoes. Agriculture and cattle rearing are the major occupations of the residents. Villagers depend on primary health centres (PHC) for their health care needs. The district has a population of 1.2 million (2011 census) with 519 villages and wards.

The district has provided MDA for LF elimination since 2004, and 12 annual rounds of MDA with DA had been completed by 2016, prior to initiation of the present study. Independent coverage surveys conducted in Yadgir after 8, 11 and 15 rounds of MDA in 2011, 2015 and 2018 showed medication consumption rates of 90.1, 56.2 and 75.4% for the eligible population [14–16]. For the remaining rounds, no data were available. Prior to the study, a total of 13 villages were surveyed for assessing Mf prevalence, jointly by the research team and the district programme to select the study sites. These villages included four sentinel and four spot-check sites (selected by the programme as a part of monitoring and evaluation in the district), and five additional sites identified based on disease prevalence. A cross sectional community survey was conducted to screen 500 individuals in each selected village with household as the sampling unit. The households were selected following a systematic sampling procedure. All the consenting individuals aged above 5 years of age in the selected households were screened for Mf using night blood sample. Mf prevalence in these thirteen sites ranged from 1.7 to 7.1% (total N surveyed, 7,589).

### Study design

This was a community-based, two-armed, open label study (S1 Protocol). Of the 13 villages, six villages were identified for the study, considering the logistics and Mf prevalence. These six villages were grouped into two blocks of 4 and 2 villages that were comparable in terms of Mf-prevalence and population size (Fig 1). The blocks were randomized for allocation of the treatment regimens. The block with Hattikuni, Gunjanur, Anpur and Nazrapur villages was

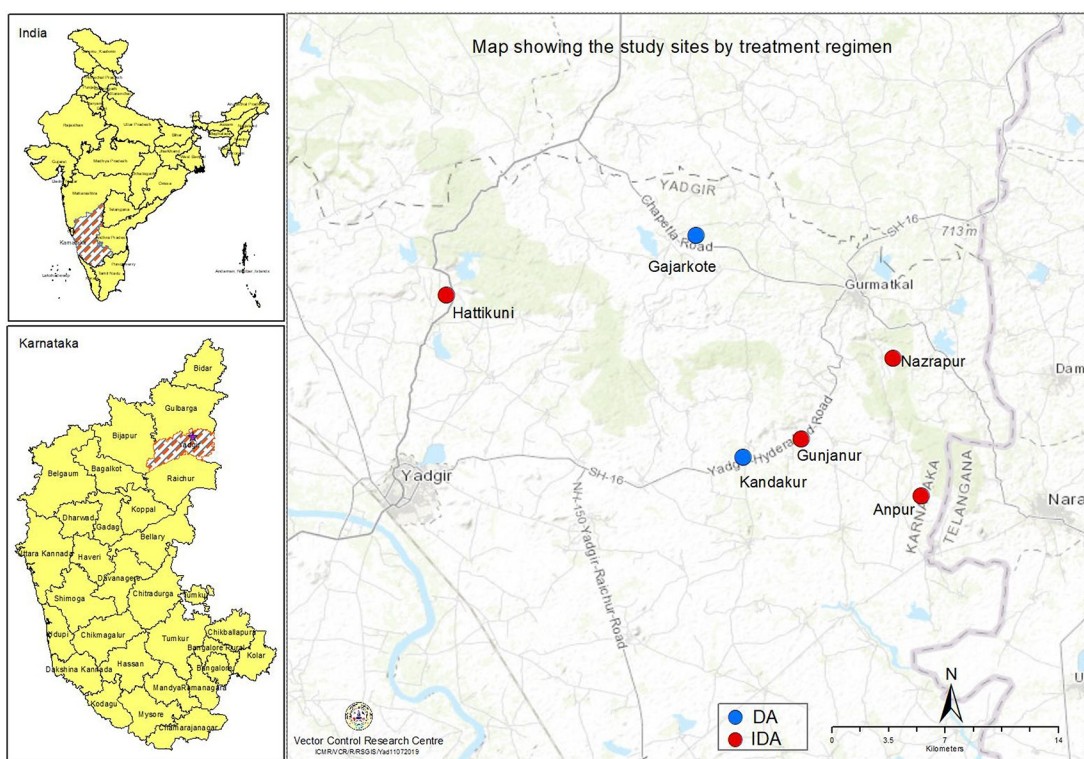

**Fig 1. Map showing the study sites by treatment regimen.**

allocated to the triple-drug regimen (IDA), and other block with Kandkur and Gajarkote villages was allocated to the double-drug regimen (DA). The IDA villages received MDA with a single dose of co-administered ivermectin, diethylcarbamazine and albendazole; villages assigned to DA received MDA with a single co-administered dose of diethycarbamazine plus albendazole (DA) without ivermectin.

## Sample size

WHO guidelines require a total sample of at least 10,000 recipients of a new treatment to provide high confidence that the rate of serious AE (SAE, the primary endpoint for safety studies) following treatment is $\leq 0.1\%$ [12]. A study with 3000 patients provides a 95% chance of identifying a single SAE if the expected incidence is 0.1% [12]. India committed to contribute at least 4000 participants for each arm to the global IDA safety study, with the balance coming from other study sites. However, the sample size for the India study alone was sufficient to demonstrate a SAE rate of less than 0.1% with 95% certainty.

A comprehensive analysis of the safety data from all the five countries that supported the WHO recommendation [3] for implementing IDA in specific situations has been published elsewhere [13]. This paper reports detailed safety results from the India study that are beyond what was reported in the global safety study paper. It also reports efficacy results that were not reported in that paper.

A separate power calculation was performed for the primary efficacy endpoint, namely complete clearance of Mf 12 months after treatment. A minimum of 35 Mf-positives in each arm was required to test whether IDA was superior to DA assuming 90% reduction in Mf-prevalence after IDA [5] and 60% reduction after DA [17] with 80% power for detecting the effect size of 30%.

## Study procedures

**Community preparation.**  Social scientists and medical social workers prepared communities by mass education and extensive communication. Medical officers and staff from PHCs, anganwadi staff (anganwadi is a childcare unit for children below 5 years, located for every 800–1000 population, run by government of India as part of Integrated Child Development Services Program to combat child hunger and malnutrition) ASHAs (Accredited Social Health Activists) and community leaders were also involved in this activity. In each village, a Community Advisory Board (CAB) was formed with six to seven members from the community for advocacy and necessary local support. Information/Education/Communication (IEC) materials such as banners, video shows and handbills were also used.

The households in the selected villages were enumerated. Enrolment teams, each with 6 members consisting of a medical officer, a junior nurse, a data entry operator (DEO) and 3 technicians, visited households in the evening hours when most family members were likely to be at home. The team identified eligible participants based on a pre-treatment health assessment using a pre-designed proforma.

**Inclusion criteria.**  Village residents aged $\geq$ 5 years and $\geq$ 15 Kg body weight who were able to provide informed consent (assent for minors) to participate, not pregnant, those who had no evidence of severe co-morbidities except for features of filarial disease and no history of previous allergy to MDA drugs were enrolled.

**Exclusion criteria.**  The study excluded persons who were (i) less than 5 years old or less than 15 Kg body weight, (ii) unable to provide informed consent (or without parental consent for minors), (iii) pregnant, (iv) women of child-bearing age who reported that their last menstrual period was 4 weeks or more prior to the study (v) women who did not recall the timing of their last menstrual period, (vi) persons with a history of severe chronic illness (chronic renal insufficiency, severe chronic liver disease, or any illness that was severe enough to interfere with activities of daily living), or (vii) persons with a known allergy to any of the study drugs.

**Participant enrolment.**  Each participant was assigned a barcode ID, and a card printed with the ID and name of the participant was handed over to the participant. The card also had phone numbers for emergency calls in case of any urgent need in the event of AEs. Matching barcode labels were affixed on FTS strips and on night blood smear microscope slides. Barcode labels were also affixed on the front wall/door of the participant's house to facilitate future follow up.

**Testing for circulating filarial antigen (CFA) and microfilaremia (Mf).**  Consenting eligible individuals were screened for CFA with a rapid format antigen test (Filariasis Test Strip or FTS, Alere Inc., Scarborough, ME, USA) in their houses. Seventy-five μl of blood was obtained by finger prick, and testing was performed as per manufacturer's instructions. Results were recorded as negative; invalid; or either weakly, moderately, or strongly positive with scores of 1, 2, or 3, respectively [18].

CFA-positive participants were screened for Mf after 9:00 PM by night blood smear examination. 60 μL of blood was collected by finger prick using a glass microcapillary tube. Dried blood smears were dehaemoglobinized, fixed and stained with Giemsa. Each slide was examined by a trained technician and later cross-examined by a second technician. A senior reader resolved any discrepancies in Mf smear results. Mf counts were expressed as the number of Mf per 60 μl of blood.

**Drug administration.**  Tablets were administered under direct observation to consenting individuals in their houses. For those negative by CFA, drugs were given immediately. For persons with positive CFA results, drugs were administered after collection of night blood smears. Care was taken to avoid drug intake on an empty stomach. Adult participants were given biscuits, and children were given biscuits or chocolates before MDA medications. Persons who vomited medications with visible tablets in the vomitus were retreated after 30 minutes.

Diethylcarbamazine dosing was age-based with a dose of 100 mg for persons aged 2–5 years, 200 mg for persons aged 6–14 years, and a maximum dose of 300 mg for persons above age 14, as per national LF elimination program guidelines in India [19] Ivermectin dosing was weight-based (200 μg/Kg body weight); the number of 3 mg tablets provided was rounded to the weight of the participant using a dosing table. Albendazole was provided with a uniform dose of 400 mg for all participants.

Ivermectin (3mg tablets) was donated by Merck Sharp and Dohme (MSD), also known as Merck & Co., (Kenilworth, NJ, USA); diethylcarbamazine (100 mg tablets; produced and donated by Eisai Co.) and albendazole (400 mg tablets; produced and donated by GlaxoSmithKline) were obtained from the National Vector Borne Diseases Control Programme (NVBDCP), Ministry of Health and Family Welfare, Government of India.

## Clinical monitoring and assessment of adverse events

**Active follow up.**   Every treated participant was visited at 24 hours and 48 hours after treatment by the same team that had enrolled them to assess and manage adverse events. A separate medical team camped in the study villages on the nights after enrolment so that they could be available to provide on-site management of early adverse events. Participants were also encouraged to call a hot line emergency cell phone number if they experienced troublesome AE.

A modified version of the document "Common Terminology Criteria for Adverse Events" (CTCAE) Version 4.03 [20] was used to categorize and score AE. Adverse events were recorded as mild (grade 1), moderate (grade 2), severe (grade 3), serious or life threatening (grade 4) and death (grade 5), and appropriate symptomatic treatment was provided. As a rough guide, grade 1 adverse events are not severe enough to interfere with the participants ability to go to school or perform work (including housework), grade 2 events interfere with school or work, and grade 3 events interfere with the participant's ability care for themselves and perform activities of daily living. Any event that poses a threat to the patient's life or that required overnight hospitalization was considered to be a serious adverse event (SAE).

**Passive monitoring.**   A separate monitoring team with a clinician, junior nurse and data entry operator visited each study village daily to passively monitor for AE on days 3–7 days post treatment; participants with AE were flagged by village coordinators for this team, and AE were recorded and managed. The passive AE assessment team also visited the PHC to record and follow participants who had reported to the centre for management of AE. PHC medical officers and staff supported the study teams for drug administration and for AE monitoring and management.

## Participant follow up for the efficacy study

For efficacy evaluations, treated participants who had positive CFA tests at baseline in both MDA treatment areas were followed up at their homes approximately one year after treatment and retested for CFA. Night blood smears were collected from CFA positive participants for detection and counting of Mf.

## Training of staff

Staff were trained on good clinical practice (GCP), use of the electronic data capture (EDC) system, clinical assessment procedures, and on testing procedures for detecting CFA and Mf. Medical teams were trained on drug administration and on assessment and management of AE prior to initiation of the study.

## Electronic data capture (EDC) system

An EDC system developed by Cliniops (Fremont, CA, USA) was used to record the data entered by data entry operators in each enrolment team. A contract research organization (CRO, Syngene International Ltd, Bengaluru, India) worked with clinical teams for data management. Each enrolment team carried one tablet computer (iPad, Apple, Cupertino, CA, USA) that was preloaded with Clinitrial software. The EDC system is 21 CFR Part 11 compliant. Title 21 CFR Part 11 is the part of Title 21 of the Code of Federal Regulations that establishes the United States Food and Drug Administration (FDA) regulations on electronic records and electronic signatures (ERES). Part 11, as it is commonly called, defines the criteria under which electronic records and electronic signatures are considered trustworthy, reliable, and equivalent to paper records (https://www.fda.gov/regulatory-information/search-fda-guidance-documents/part-11-electronic-records-electronic-signatures-scope-and-application accessed 27 August 2020). Electronic case report forms (eCRF) were developed to comply with International Council for Harmonization on Good Clinical Practice (ICH GCP) and CDASH/CDISC standards. Barcode ID and demographic/clinical details of each participant as well as CFA and Mf test results were entered into the eCRF. Data from each iPad was uploaded to an online server on the same day they were collected. Syngene closely monitored the study activities including data entry, data uploading, and data integrity with help from a data manager at Washington University in St. Louis who generated queries that were addressed onsite in India.

## Data analysis

**Age and gender distribution between treatment areas.**   We used a mixed effect linear regression model that assumed a normal distribution with an identity link function to compare differences by age [children (5–17 years) and adults (≥18 years)], and a mixed effect logistic regression model with logit link function to compare the difference in sex-ratio between treatment groups. For both analyses, village (cluster) was used as a random effect.

**Association of AE with demographic, filarial infection and drug regimen.**   The association of AE occurrence with drug regimen (DA or IDA), age (adults or children), gender and infection status (Mf or CFA) was examined using the chi-squared test. While assessing the infection status, comparisons were made between those who were both Mf and CFA negative, Mf +ve and CFA +ve, and Mf-ve but CFA +ve, separately for children and adults and for each MDA treatment regimen.

A generalized linear mixed modelling approach was used to assess the association of occurrence of AE with the above-mentioned factors by including clusters as random effect. Occurrence of AE was assumed to follow a binomial probability distribution with logit link function. The treatment effect was adjusted by clusters, Mf and/or CFA test positivity, by age group (children and adults), and gender. The significance of the random effects was assessed using the likelihood ratio test of the models with and without random effects.

**Drug efficacy.**   The efficacy of drug regimens was assessed in terms of (i) clearance or persistence of CFA and (ii) clearance/reduction of Mf relative to baseline levels. Persons who were tested negative for CFA were assumed to be Mf negative, as CFA is an indication of live adult worms, in the absence of which no Mf can be expected. Therefore, the denominator for the clearance of Mf is all the persons negative for CFA plus the number of CFA-positive persons who were screened for Mf.

The chi-squared test was used to compare the reductions in infections (Mf or CFA) between treatment areas. Logistic regression analysis was also used to assess the association of Mf or CFA clearance 1-year post treatment, with drug regimen, age group (children or adults) and gender. Clearance of Mf or CFA was assumed to follow a binomial distribution with logit

link function. Separate analyses were performed for Mf and CFA clearance. For the Mf clearance analysis, the treatment effect was adjusted using age group (children or adults), gender and Mf-count at baseline as confounders. For the CFA clearance analysis, in addition to age (children or adults) and gender, an interaction term "Drug regimen × FTS-score at baseline" was included to account for FTS-score at baseline as an effect modifier.

Among the participants who were positive for CFA or Mf at baseline, percentage reductions in FTS score or Mf counts one-year post-MDA relative to baseline levels were compared by treatment arm using the Mann-Whitney U-test. This analysis was also performed separately for children, adults and each sex. The Pearson correlation coefficient was used to assess the likelihood of participants becoming amicrofilaremic after treatment from baseline Mf counts. Analysis of covariance (ANCOVA) was used to compare the reductions in Mf counts between drug regimens, with Mf-count one-year post-MDA as a dependent variable and baseline Mf-count, age (children or adults) and sex of the subjects as independent variables. The Mf-count was log-transformed after adding 1 to each number, and data are presented as geometric means with 95% CIs. The ANCOVA was performed on the log-transformed data. All statistical analyses were carried out using the statistical software package STATA version 14.2 (StataCorp LLC, College Station, TX, USA).

## Results

### Enrolment and demographics

A CONSORT diagram showing the details of treatment allocation and AE follow-up is provided in Fig 2. A total of 4758 and 4160 consenting participants were enrolled and treated in the triple and double-drug treatment areas, respectively, from October 2016 to March 2017 (Table 1). Among the eligible population, 73.5% received treatment in the IDA treatment areas and 82.9% in DA treatment areas. Treated populations between areas were comparable with regards to demographic characteristics (mean age and gender) (P>0.4 for both comparisons). Further, both mean age and sex ratio did not vary significantly between villages (the random intercepts were not significantly different from zero, P>0.1 for both models).

### Prevalence of filarial infection at baseline

CFA prevalence was 26.4% in the IDA treatment area and 24.5% in the DA treatment area (Table 2). CFA prevalence was significantly higher in males than in females in both treatment areas. CFA prevalence in females was significantly higher in the IDA treatment area than in the DA area. Distributions of FTS scores were similar in the two treatment areas.

Baseline Mf prevalence and geometric mean intensity (GMI, Mf counts) for all participants and geometric mean density (among Mf-positives) were similar in the two treatment areas. Mf-prevalences were significantly higher in males than in females in both treatment area.

CFA and Mf prevalence data are shown separately for children (ages 5–17 years) and adults (≥18 years) in Table 3. Prevalences for both infection parameters were much higher in adults than in children. Baseline Mf prevalence was significantly higher in children in the IDA treatment area compared to that in the DA treatment area.

### AE assessment and AE rates following treatment

The participant follow-up rates for AE assessment were very high and comparable in both treatment areas (S1 Table). However, the vast majority of AE in both treatments (IDA vs DA) were mild (Grade 1: 90.5 vs 97.7%) or moderate (Grade 2: 9.3 vs 2.4%). One participant who received IDA had severe diarrhoea on day 4 post-treatment (a Grade 3 AE: 0.2%) that resolved by day 6. Diarrhoea is common in India, and this AE may or may not have been related to

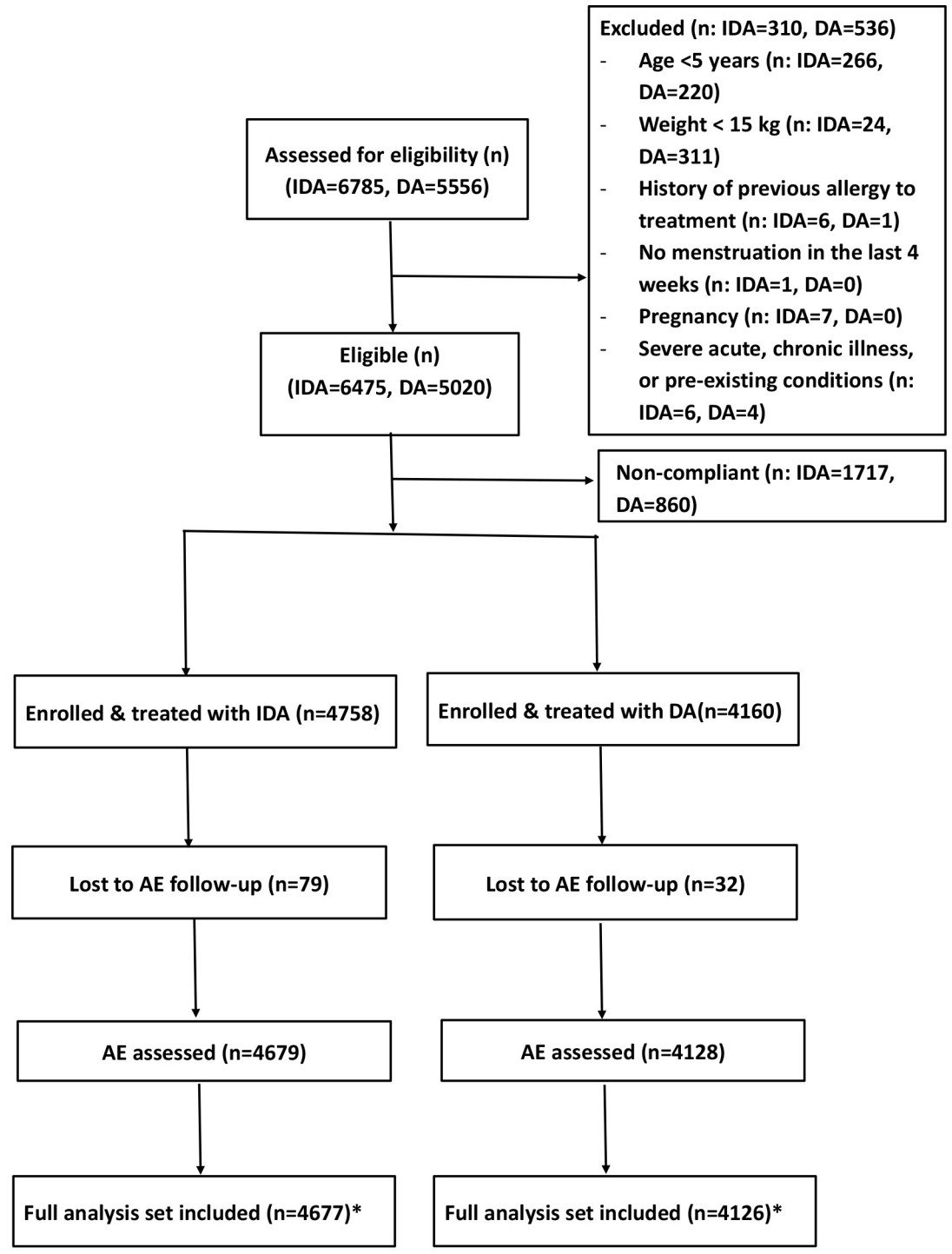

**Fig 2. CONSORT diagram.** Treatment allocation was by block randomization (DA: Diethyl carbamazine plus albendazole and IDA: ivermectin+DA).

**Table 1. Study villages, treatment regimen and demographic features of the study population.**

| Village | Study arm | Number treated | Mean age[a] (SD) in years | Female (%)[b] |
|---|---|---|---|---|
| Hattikuni | IDA | 2045 | 26.4 (16.1) | 1029 (50.4) |
| Gunjanur | IDA | 714 | 25.8 (15.8) | 370 (51.8) |
| Anpur | IDA | 1424 | 26.4 (14.8) | 736 (51.7) |
| Nazrapur | IDA | 575 | 27.4 (15.0) | 291 (50.6) |
| Overall IDA | | 4758 | 26.4 (15.6) | 2426 (51.0) |
| Kandkur | DA | 1419 | 27.1 (16.2) | 761 (53.7) |
| Gajarkote | DA | 2741 | 25.6 (15.5) | 1392 (50.8) |
| Overall DA | | 4160 | 26.1 (15.8) | 2153 (51.8) |

[a] Mean age did not differ significantly between treatment areas (P = 0.61, random effects linear regression model with village as random effect).

[b] Sex ratio did not differ significantly between treatment areas (P = 0.47, random effects logistic regression model with village as random effect).

treatment. There were no serious adverse events (Grades 4 or 5) recorded after either treatment.

## Risk factors for AE

AEs were more frequent among females than males in each treatment area. AEs were significantly more common after IDA than after DA in males but not in females. AEs were significantly more frequent in adults than in children (Table 4).

AEs were slightly but significantly more frequent after IDA treatment than after DA (Table 5). Table 5 also shows AE rates by Mf status and treatment arm. AE rates were higher in microfilaremic participants in both treatment areas. Microfilaremic participants had higher AE rates after IDA than after DA, but AE rates after IDA and DA were similar in persons without Mf. Most AE were mild (Grade 1) after IDA or DA regardless of Mf status. Pre-treatment

**Table 2. Filarial infection status of the study population prior to treatment.**

| | IDA | DA | P value |
|---|---|---|---|
| **CFA positive (%)[a]** | | | |
| Female (n = 2151 DA; 2414 IDA) | 608 (25.2) | 470 (21.9) | 0.01 |
| Male (n = 2004 DA; 2318 IDA) | 648 (27.8) | 549 (27.4) | 0.75 |
| **Both gender (n = 4155 DA, 4732 IDA)** | **1256 (26.4)** | **1019 (24.5)** | **0.04** |
| **FTS scores[a]** | | | |
| Weak | 189 (4.0) | 148 (3.6) | 0.31 |
| Medium | 323 (6.8) | 241 (5.8) | 0.05 |
| Strong | 744 (15.7) | 630 (15.2) | 0.52 |
| **Microfilaremia positive (%)[b]** | | | |
| Female (n = 2132 DA; 2401 IDA) | 141 (5.9) | 115 (5.4) | 0.48 |
| Male (n = 1987 DA; 2305 IDA) | 185 (8.0) | 150 (7.5) | 0.56 |
| **Both genders (n = 4119 DA, 4706 IDA)** | **326 (6.9%)** | **265 (6.4%)** | **0.35** |
| Geometric mean Mf intensity per 60 μl among all persons | 0.19 | 0.16 | 0.09 |
| Geometric mean Mf density per 60 μl in Mf-positive persons | 10.46 | 8.49 | 0.11 |

[a] Denominator excludes persons with missing (7) or indeterminate circulating filarial antigenemia (CFA) test results (n = 20)

[b] Denominator excludes persons who were missing or refused to participate and includes CFA negatives (not tested for Mf which were assumed to be negative for Mf) and CFA positives tested for Mf.

**Table 3. Age-specific prevalence of CFA and Mf by MDA treatment areas.**

| Age-class (Years) | No. screened for CFA | | No. positive for CFA (%) | | No. of people with night blood testing | | No. positive for Mf (%)* | |
|---|---|---|---|---|---|---|---|---|
| | IDA | DA | IDA | DA | IDA | DA | IDA | DA |
| Children (5–17 years) | 2756 | 1529 | 622 (22.6) | 264 (17.3) | 615 | 246 | 175 (6.4) | 41 (2.7) |
| Adults (≥18 years) | 1976 | 2626 | 634 (32.1) | 755 (28.8) | 615 | 737 | 151 (7.7) | 224 (8.6) |
| Overall# | 4732 | 4155 | 1256 (26.5) | 1019 (24.5) | 1230 | 983 | 326 (6.9) | 265 (6.4) |

*Denominator includes CFA negatives (not tested for Mf which were assumed to be negative for Mf) and CFA positives tested for Mf.

#Excludes persons with indeterminate circulating filarial antigenemia (CFA) test results (n = 20) and missing FTS results (7).

Prevalences of circulating filarial antigenemia (CFA) differ significantly between treatment areas [P<0.05 for children (5–17 yr) and adults].

Prevalences of Mf (microfilaremia) differ significantly between treatment areas (P<0.001) for children (5–17 yr) but the difference was not significant for adults (P = 0.29).

Mf counts were significantly higher in persons who experienced AE after IDA (geometric means: 15.8 [95% CI:10.3–24.4] vs 5.9 [4.0–8.7]) or DA (18.2 [12.5–26.3] vs 7.0 [5.8–8.5]) (P<0.001 for both comparisons).

Table 6 shows AE rates by pre-treatment CFA status and treatment area. AE were more common among CFA-positive participants than in CFA-negatives after IDA or DA treatment. AE rates were also higher in CFA-positive participants after IDA than after DA. Again, the

**Table 4. Adverse events by age-group, gender and treatment regimen.**

| Treatment group | Gender | Age-class (Years) | No. assessed for AEs | No. with AE (%) | No. with Grade 1 AE (%) | No. with Grade 2 AE (%) | No. with Grade 3 AE (%) |
|---|---|---|---|---|---|---|---|
| IDA | Female | 5–17 | 804 | 45 (5.6) | 41 (5.1) | 4 (0.5) | 0 (0.00) |
| | | > = 18 | 1596 | 179 (11.2) | 158 (9.9) | 21 (1.3) | 0 (0.00) |
| | | **Overall** | **2400** | **224 (9.3)** | **199 (8.3)** | **25 (1.0)** | **0 (0.00)** |
| | Male | 5–17 | 928 | 52 (5.6) | 48 (5.1) | 4 (0.4) | 0 (0.00) |
| | | > = 18 | 1351 | 112 (8.3) | 104 (7.7) | 7 (0.5) | 1 (0.01) |
| | | **Overall** | **2278** | **164 (7.2)** | **152 (6.7)** | **11 (0.5)** | **1 (0.04)** |
| | **Both** | 5–17 | 1732 | 97 (5.6) | 89 (5.1) | 8 (0.5) | 0 (0.00) |
| | | > = 18 | 2947 | 291 (9.9) | 262 (8.9) | 28 (0.9) | 1(0.03) |
| | | **Overall** | **4679** | **388 (8.3)** | **351 (7.5)** | **36 (0.8)** | **1 (0.02)** |
| DA | Female | 5–17 | 743 | 41 (5.5) | 39 (5.2) | 2 (0.3) | 0 (0.00) |
| | | > = 18 | 1393 | 140 (10.1) | 137 (9.8) | 3 (0.2) | 0 (0.00) |
| | | **Overall** | **2136** | **181 (8.5)** | **176 (8.2)** | **5 (0.2)** | **0 (0.00)** |
| | Male | 5–17 | 776 | 24 (3.1) | 24 (3.1) | 0 (0.0) | 0 (0.00) |
| | | > = 18 | 1216 | 58 (4.8) | 57 (4.7) | 1 (0.1) | 0 (0.00) |
| | | **Overall** | **1992** | **82 (4.1)** | **81 (4.1)** | **1 (0.1)** | **0 (0.00)** |
| | **Both** | 5–17 | 1519 | 65 (4.3) | 63 (4.1) | 2 (0.1) | 0 (0.00) |
| | | > = 18 | 2609 | 198 (7.6) | 194 (7.4) | 4 (0.2) | 0 (0.00) |
| | | **Overall** | **4128** | **263 (6.4)** | **257 (6.2)** | **6 (0.1)** | **0 (0.00)** |

The overall AE rates and severity of AE differ by treatment regimens (P<0.01 for all comparisons).

In IDA and DA treatment areas, AE rates (Overall, Grade 1 and Grade2) were significantly different between females and males (p<0.05).

AE rates (overall and Grade 1) did not different significantly among females between IDA and DA treatment areas (P>0.05).

Grade 2 AE was significantly higher in IDA arm compared to DA arm (P<0.05).

AE rates (overall, grade 1 and grade 2) differ significantly between treatment areas among males (p<0.001).

AE rate significantly higher after IDA than after DA in adult (P = 0.003), but the difference was not significant for children (P = 0.084).

**Table 5. Adverse events by microfilaremia (Mf) and treatment regimen.** No AE's beyond Grade 3 were observed.

| Drug regimen | Mf test results | Total assessed for AE*s | Any AE (%) | Grade 1 AE (%) | Grade 2 AE (%) | Grade 3 AE (%) |
|---|---|---|---|---|---|---|
| IDA (n = 4627) | Mf(-) | 4311 | 254 (5.9) | 241 (5.6) | 18 (0.4) | 1 (0.02) |
| | Mf(+) | 316 | 128 (40.5) | 110 (34.8) | 18 (5.7) | 0 (0.0) |
| DA (n = 4087) | Mf(-) | 3825 | 209 (5.5) | 204 (5.3) | 6 (0.2) | 0 (0.0) |
| | Mf(+) | 262 | 53 (20.2) | 53 (20.2) | 0 (0.0) | 0 (0.0) |

* Excludes treated persons with missing Mf and those not available for assessing AE following treatment.

Among Mf negatives, only grade 2 AE was significantly different between IDA and DA treatment areas (p = 0.03).

Among Mf positives, AE rates (overall and, Grades 1 and 2) differed significantly between IDA and DA treatment areas (p<0.001).

AE rates were more common in Mf positives than in Mf-negatives in both treatment areas (P<0.05 for both comparisons).

vast majority of AE were mild (grade 1). AE rates in CFA-negatives were similar after IDA or DA. The rate of grade 2 AE among CFA negatives after IDA (0.4%) was slightly but significantly higher than that after DA (0.1%) (P = 0.008).

Multivariate logistic regression analysis showed that after adjusting for age, gender and infection status, the occurrence of AE did not differ significantly between IDA and DA. However, infection status (Mf and/or Ag status), gender, and age were all significantly associated with AE after either treatment (Fig 3). The odds of microfilaremic children experiencing an AE were almost 32 (95%CI: 17.8–38.5) and 14.4 (7.2–19.1) times higher relative to children with negative Mf and CFA tests (uninfected), after IDA or DA, respectively. The odds of AE in microfilaremic adults, were 10.2 (7.5–16.6) and 3.7 (2.5–6.4) times higher after IDA or DA treatment than in uninfected participants. In contrast, the odds ratio for AE in persons with isolated CFA positivity (amicrofilaremic subjects) were only 1.9 (1.4–3.1) after IDA and 1.4 (0.9–2.3) after DA in adults, and 1.5 (0.8–1.9) and 2.1 (1.2–2.5) in children after IDA and DA respectively. The multivariate analysis confirmed the increased risks for AE in females and in adults that were observed in the univariate analysis.

The odds of AE after IDA in infected children (CFA positive, with or without Mf) was not significantly different from that of infected children after DA (the 95% CIs overlap). However, the odds of AE in infected adults was significantly higher after IDA than after DA (the 95% CIs do not overlap). A similar comparison for adults with isolated CFA without Mf showed that the odds of AE were comparable after IDA or DA (the 95% CIs overlap, Fig 3).

**Table 6. Adverse events by circulating filarial antigenemia (CFA) status and treatment regimens.** No AE's beyond Grade 3 were observed.

| Drug regimen | CFA | No. assessed for AEs* | Any AE (%) | Grade 1 AE (%) | Grade 2 AE (%) | Grade 3 AE (%) |
|---|---|---|---|---|---|---|
| IDA (n = 4677) | Positive | 1236 | 214 (17.3) | 192 (15.5) | 21 (1.7) | 1 (0.1) |
| | Negative | 3421 | 172 (5.0) | 157 (4.6) | 15 (0.4) | 0 (0.0) |
| | Undetermined | 20 | 2 (10.0) | 2 (10.0) | 0 (0.0) | 0 (0.0) |
| DA (n = 4126) | Positive | 1012 | 104 (10.6) | 104 (10.3) | 3 (0.3) | 0 (0.0) |
| | Negative | 3111 | 153 (5.0) | 153 (4.9) | 3 (0.1) | 0 (0.0) |
| | Undetermined | 3 | 0 (0.0) | 0 (0.0) | 0 (0.0) | 0 (0.0) |

* Excludes treated persons with missing CFA TEST results and those not available for assessing AE following treatment.

Among CFA positives, AE rates (overall and grades 1 and 2) differ significantly between IDA and DA treatment areas (p<0.001).

Among CFA negatives, only grade-2 AE was significantly different between IDA and DA treatment areas (p = 0.008).

AE rates among undetermined did not differ significantly between IDA and DA treatment areas (p>0.05).

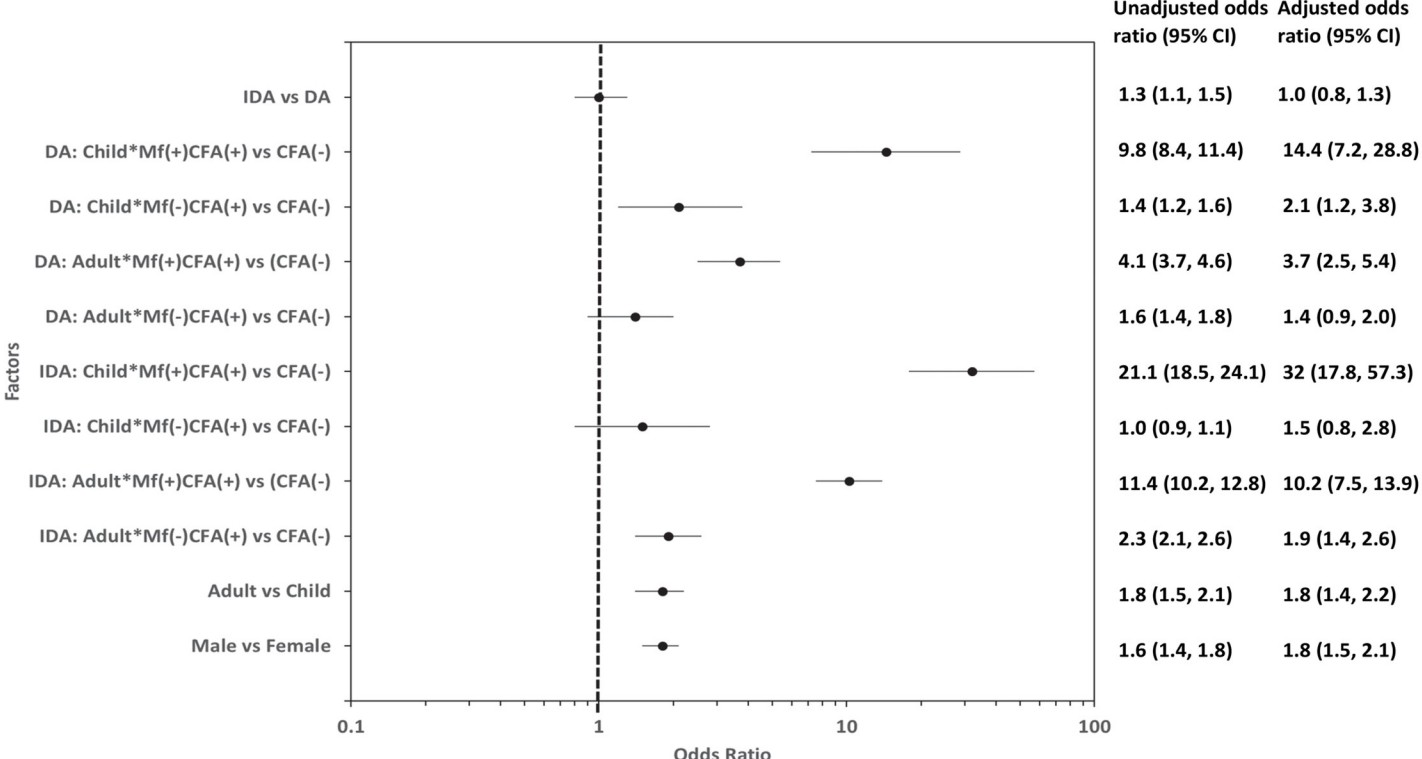

**Fig 3. Forest plot showing unadjusted and adjusted odds ratios for factors associated with AE following MDA for filariasis.**

## Types of adverse events recorded after treatment

The most common AEs reported after IDA or DA treatment were fever (IDA vs DA: 23.0 vs 13.4%), headache (22.2 vs 22.2%), dizziness (11.2 vs 13.9%), and gastrointestinal symptoms (nausea or vomiting: 13.3 vs 17.7%, S2 Table). Of the most common AEs, fever was significantly higher after IDA than DA (P<0.0002). Mild fever (Grade 1) was more prominent after IDA (19.9%) and mild headache (Grade 1) after DA (21.7%).

In participants with post-treatment AE assessments, fever (2.7 vs 0.7%) and headache (2.5 vs 1.6%) were more frequent after IDA (n = 4679) than after DA (n = 4128). Frequencies for all other types of AE were similar after IDA and DA (Fig 4). Fever, headache, fatigue, dizziness, muscle pain, vomiting and nausea were more common after IDA than after DA in persons with Mf or positive CFA tests (many of whom had Mf (S1 and S2 Figs). Fever and headache were the most common AE recorded for microfilaremic persons after DA. Besides fever and headache, dizziness, muscle and joint pain were all more common after DA treatment in persons with positive CFA tests (S1 and S2 Figs).

A few participants (6 after IDA and 2 after DA) developed localized reactions which include soft tissue swellings, scrotal swelling and pain, lymph node swellings and pain and lower limb edema indicating death of adult worms [21]. All these events were also self-limited and responded to treatment with mild analgesics with or without non-steroidal anti-inflammatory agents.

## Drug efficacy

**Mf-clearance.** A CONSORT diagram with information on follow-up testing one year after MDA with either IDA or DA for persons with positive tests for filarial infection at

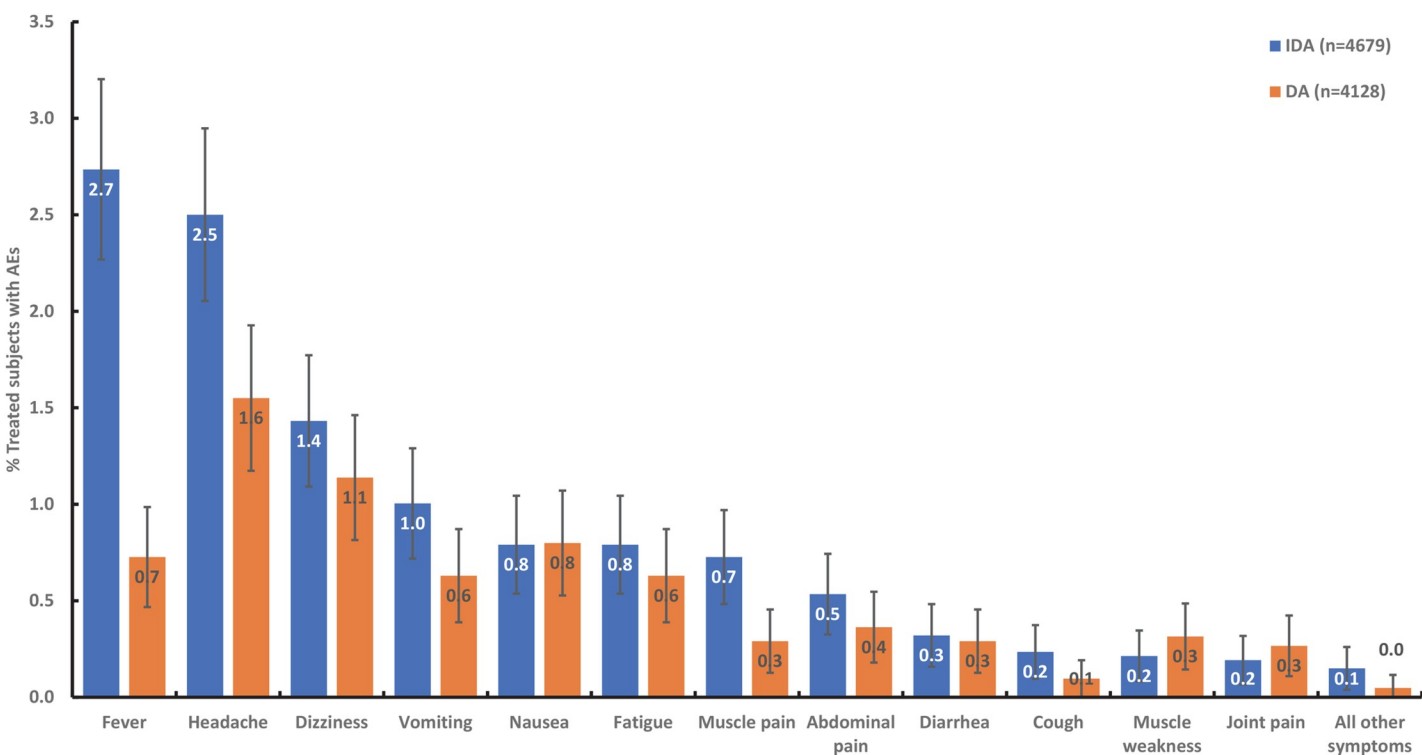

**Fig 4. No. of persons with the most commonly observed AE by treatment regimen expressed as percentages of participants who were assessed for AE after treatment.** A participant was counted only once for each AE type (e.g., a subject can only have a single headache). But if a participant experienced different AE types he/she was included in the numerator of each AE type (e.g., if a subject experience a headache and fatigue, then they will be included in the numerator for both of these AE categories). AE, adverse event; DA, double-drug therapy (diethylcarbamazine, albendazole); IDA, triple-drug therapy (ivermectin, diethylcarbamazine, albendazole).

baseline is shown in Fig 5. We were able to retest 268 of 324 Mf positive persons (82.7%) after IDA and 212 out of 264 (80.3%) after DA. Complete clearance of Mf was observed in 84.0% of the Mf positive participants after IDA, and this was significantly higher than the 61.8% Mf clearance rate after DA (P<0.001) (Table 7). Mf-clearance rates were significantly higher after IDA in both females and males than after DA. The same trend was observed in children, but relatively few children were infected, and the difference in treatment efficacy was not statistically significant. A multivariate logistic regression analysis confirmed that the percentage of participants who cleared Mf was significantly higher after IDA than DA [OR (95% CI): 3.3 (2.1–5.1)] after adjusting for the effects of age, gender and Mf-count at baseline.

The likelihood of individuals becoming amicrofilaremic after DA (range: 21.4–100% for varying levels of Mf counts at baseline) was negatively associated with pre-treatment Mf-counts (Fig 6, r = -0.98, P<0.005). However, total Mf clearance after IDA (range: 75.0–93% for varying levels of Mf counts at base line) was independent of pre-treatment Mf-counts (Fig 6, r = -0.54, P = 0.16). The two treatments were almost equally effective for clearing Mf when baseline counts were low. However, the Mf clearance rates were lower after DA relative to those treated with IDA in persons with higher baseline Mf counts (greater than 32 Mf per 60 μl of night blood).

**Mf-density reduction.** The pre-treatment geometric mean Mf-densities (GMf) in the retested Mf positive cohort persons were 11.8 per 60 μl (range 1–712) and 9.5 (1–835) in IDA and DA treatment areas, respectively. The GMf values decreased to 0.40 (0–228) after IDA and 1.0 (0–150) after DA (Table 7) at 12 months post treatment. IDA was more effective (96.4%)

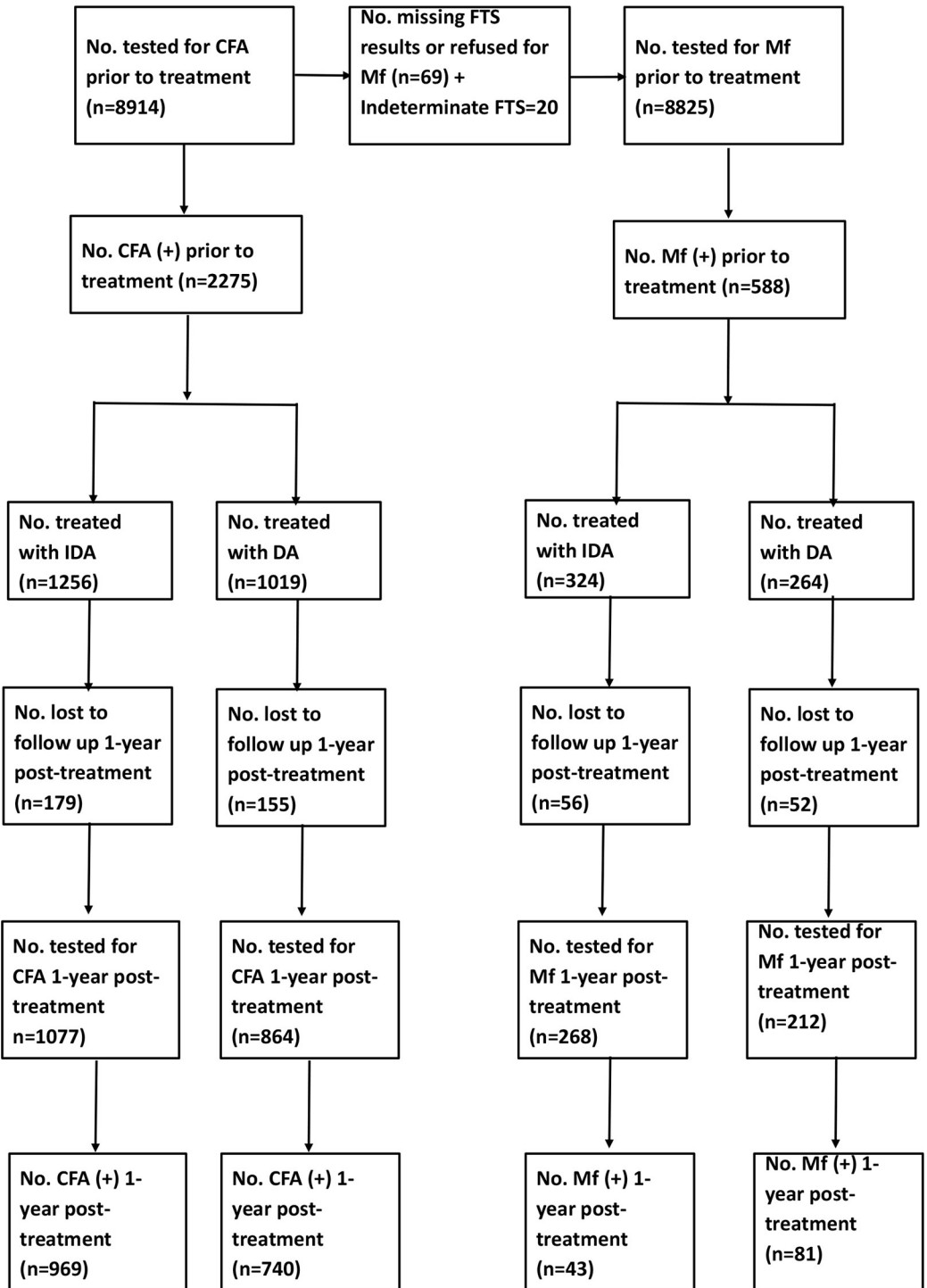

**Fig 5. CONSORT diagram.** Information on individuals with filarial infections at baseline and who were retested and retreated 1-year post-treatment.

for reducing Mf counts than DA (90.0%). An analysis of covariance (ANCOVA) showed that the reduction in GMf density was significantly greater after IDA than after DA after adjusting for age, gender and baseline Mf count (P< 0.002).

**Table 7. Reductions in microfilaria (Mf) positivity and geometric mean Mf-count 1-year post-treatment with IDA or DA.** Reduction in Mf-positivity is calculated as % becoming negative for Mf among persons Mf-positive prior to treatment.

| Drug regimen | Gender | Age-class (Years) | *No. Mf (+) prior to treatment | No. followed up for Mf (%) | No. positive for Mf (%) | % reduction 1-year post-treatment | Geometric mean Mfc per 60 μl (range) | | |
|---|---|---|---|---|---|---|---|---|---|
| | | | | | | | Prior to treatment | #1-year post-treatment | % Reduction 1-year post-treatment |
| **IDA** | Female | 5–17 | 28 | 20 (71.4) | 4 (20.0) | 80.0 | 17.3 (2–712) | 0.5 (0–14) | 97.2 |
| | | > = 18 | 112 | 96 (85.7) | 12 (12.5) | 87.5 | 11.2 (1–289) | 0.4 (0–228) | 96.7 |
| | | **Overall** | **140** | **116 (82.9)** | **16 (13.8)** | **86.2** | **12.2 (1–712)** | **0.4 (0–228)** | **96.8** |
| | Male | 5–17 | 28 | 18 (64.3) | 5 (27.8) | 72.2 | 10.6 (1–215) | 0.5 (0–6) | 95.6 |
| | | > = 18 | 156 | 134 (85.9) | 22 (16.4) | 83.6 | 11.6 (1–252) | 0.5 (0–220) | 96.1 |
| | | **Overall** | **184** | **152 (82.6)** | **27 (17.8)** | **82.2** | **11.4 (1–252)** | **0.5 (0–220)** | **96.0** |
| | **Both** | 5–17 | 56 | 38 (67.9) | 9 (23.7) | 76.3 | 13.6 (1–712) | 0.5 (0–14) | 96.5 |
| | | > = 18 | 268 | 230 (85.8) | 34 (14.8) | 85.2 | 11.4 (1–289) | 0.4 (0–228) | 96.3 |
| | | **Overall** | **324** | **268 (82.7)** | **43 (16.0)** | **84.0** | **11.8 (1–712)** | **0.4 (0–228)** | **96.4** |
| **DA** | Female | 5–17 | 23 | 18 (78.3) | 5 (27.8) | 72.2 | 13.1 (1–105) | 0.8 (0–21) | 93.7 |
| | | > = 18 | 92 | 76 (82.6) | 36 (47.4) | 52.6 | 12.2 (1–180) | 1.2 (0–150) | 89.9 |
| | | **Overall** | **115** | **94 (81.7)** | **41 (43.6)** | **56.4** | **12.3 (1–180)** | **1.1 (0–150)** | **90.7** |
| | Male | 5–17 | 18 | 14 (77.8) | 7 (50.0) | 50.0 | 10.4 (1–304) | 2.0 (0–93) | 80.4 |
| | | > = 18 | 131 | 104 (79.4) | 33 (31.7) | 68.3 | 7.4 (1–835) | 0.7 (0–67) | 90.7 |
| | | **Overall** | **149** | **118 (79.2)** | **40 (33.9)** | **66.1** | **7.7 (1–835)** | **0.8 (0–93)** | **89.5** |
| | **Both** | 5–17 | 41 | 32 (78.0) | 12 (37.5) | 62.5 | 11.8 (1–304) | 1.3 (0–93) | 89.2 |
| | | > = 18 | 223 | 180 (80.7) | 69 (38.3) | 61.7 | 9.1 (1–835) | 0.9 (0–150) | 90.1 |
| | | **Overall** | **264** | **212 (80.3)** | **81 (38.2)** | **61.8** | **9.5 (1–835)** | **1.0 (0–150)** | **90.0** |

* **Excludes individuals aged < 5 years old, 2 individuals each in IDA and DA for whom age or gender not available.**

# **GMf values are for persons who were Mf-positive at baseline and then retested 1-year post-treatment.**

**Statistical comparison for % reduction in Mf prevalence, IDA vs DA.**

**Female:** P value: 0.0001; Age-class: 0.57 (5–17 yrs); 0.0001 (≥18 yrs).

**Male:** P value: 0.002; Age-class: 0.19 (5–17 yrs); 0.005 (≥18 yrs).

**Both gender:** P value: 0.21 (5–17 yrs); 0001 (≥18 yrs).

**Overall:** P value: 0.0001.

**Statistical comparison for % reduction in Mf-density, IDA vs DA.**

**Female:** P value: 0.0645; 0.28 (5–17 yrs); 0.1186 (≥18 yrs).

**Male:** P value: 0.0007; 0.98 (5–17 yrs); 0.0003 (≥18 yrs).

**Both gender:** P value:0.3917 (5–17 yrs); 0.0002 (≥18 yrs).

**Overall:** P value: 0.0002.

**CFA-clearance.** Neither IDA nor DA were very effective for clearing CFA; 1077 of 1256 CFA positives (85.7%) in the IDA arm and 864 of 1019 CFA-positives (84.8%) in the DA arm were retested 1-year post treatment. CFA clearance rates were 10% after IDA and 14.4% after DA; this difference was statistically significant (P = 003) (S3 Table). Gender wise analysis showed that this difference in CFA clearance between treatments was significant in males only.

CFA clearance was more common in persons with low FTS scores of "1" at baseline. While CFA clearance rates were comparable between IDA and DA for people with baseline FTS scores of 1 or 2 (P>0.15 for both comparisons), clearance rates were higher after DA in persons with baseline FTS scores of 3 (P = 0.0005).

A multivariate logistic regression analysis showed that CFA clearance was comparable between DA and IDA after adjusting for age and gender (OR [95% CI] = 0.79 [0.48–1.3]; P = 0.35). However, the CFA clearance rates were significantly higher in males and lower in

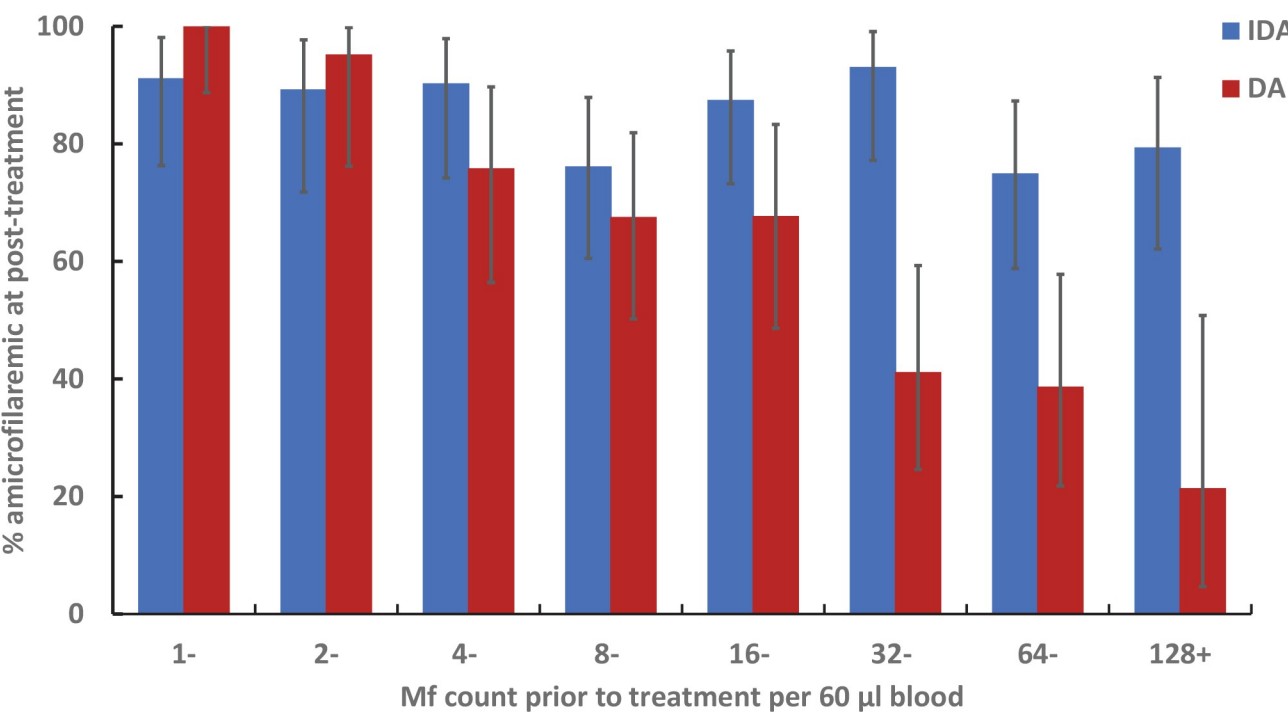

**Fig 6. Distribution of subjects who were amicrofilaremic 1 year after treatment by pre-treatment microfilaria count.** Error bars 95% CI based on exact binomial probability distribution.

adults compared to those in females and children, respectively, after both treatments. Complete CFA clearance rates decreased with increasing FTS baseline scores after both treatments (S3 Table). Some of the participants with isolated CFA positivity at baseline (IDA: 3.2%, n = 691; DA: 5.0%, n = 519) were Mf-positive 12-months after treatment, but this difference was not significant. None of 53 children aged 5–9 years in this study with isolated CFA positivity was Mf positive one year after IDA. In contrast, 2 (4.7%) out of 42 children with isolated CFA positivity at baseline were Mf-positive 12 months after DA.

Mean FTS scores were comparable between pre- and post-treatment with IDA in children (2.39 vs 2.03), adults (2.46 vs 2.25) and in all participants (2.44 vs 2.19). Similar results were observed for changes in scores in children (2.46 vs 1.97), adults (2.49 vs 2.22) and in all participants (2.48 vs 2.16) after DA. The changes in FTS scores between pre and 12 months post-treatment were not significant in the corresponding age-groups as well as between drug regimens after DA or IDA (P>0.14, for all comparisons).

Figs 7 and 8 compare the percentage of CFA positives tested positive for Mf at baseline and 12-months after IDA and DA treatments respectively. While at baseline, the percentage was comparable between drug regimens (P>0.05), it declined significantly after IDA at 12 months post-treatment (P<0.05). The percentages were significantly lower in both children and adults after IDA compared with baseline. However, a similar comparison showed that it was significant only in adults after DA (P<0.005, for all comparisons).

## Discussion

This study contributed one-third of the 12280 total number of participants that were treated with IDA in the multicentre IDA global safety study [13], the results of which helped to fulfil requirements for policy change at WHO regarding MDA for LF elimination [3]. Both CFA

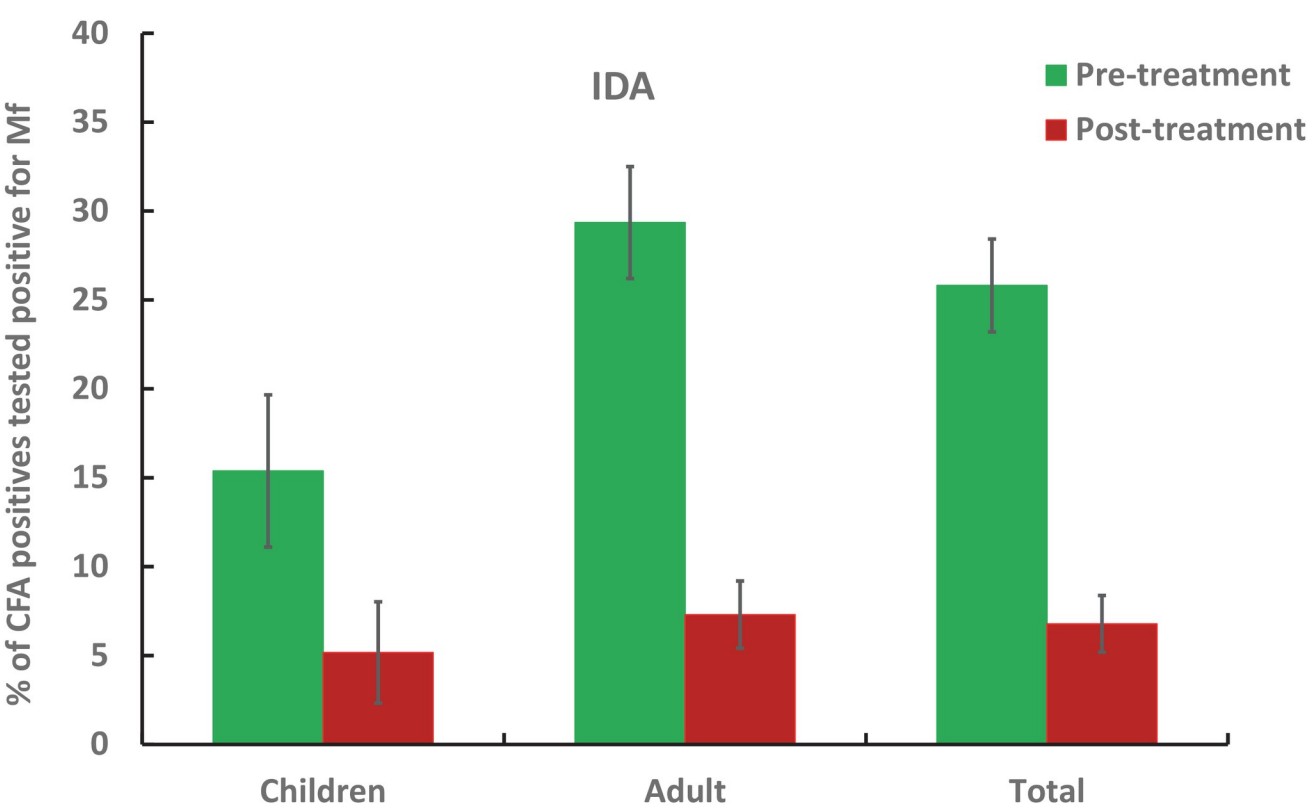

**Fig 7. Percentage of CFA positives tested positive for Mf prior to and 1-year post treatment with IDA in children, adult and for all ages.**

and Mf rates at baseline were higher in the India study site compared to other sites that participated in the global safety study (except for one of the districts studied in Fiji) [13]. The India study found that Mf prevalence was higher in males, and this was also observed in all of the other safety study sites. The baseline geometric mean Mf count in microfilaremic individuals in this study was comparable to that reported from Fiji [22].

The comparative AE results from our study were generally consistent with those of the multicentre global study [13]. The overall AE rates in the global study after IDA and DA were about 12% for each. Although our study showed a slightly higher frequency of AE after IDA than DA, multivariate analysis (after adjusting for age, gender and filarial infection) showed that the difference was not significant. Adults, females, and LF infected persons were at higher risk for AEs. Despite the relatively high infection prevalence in our study areas, AEs were relatively low after IDA (8.3%) or DA (6.4%) compared to other sites [13]. In contrast to safety study results from Haiti and Fiji, there were no serious adverse events after treatment in the India study. This could be due to lower Mf densities in the India study area or to the treatment procedures followed in our study. Participant enrolment was done in the evening when most family members were present. Those who were enrolled in the early evening were given biscuits before treatment and advised to have an early dinner. Most of those enrolled in the late evening had their dinner before treatment. This ensured that no treatment was given on an empty stomach. This could be the reason why AEs such as fatigue, dizziness and nausea were minimal in this study in contrast to Fiji, where fatigue constituted 8.2% of AEs. Fever and headache together accounted for 45% (IDA, n = 623) and 35% (DA, n = 396) of total AE in this study. These AE are considered to be more associated with treatment of filarial infections than gastrointestinal AEs. Timely active detection and management could have prevented a

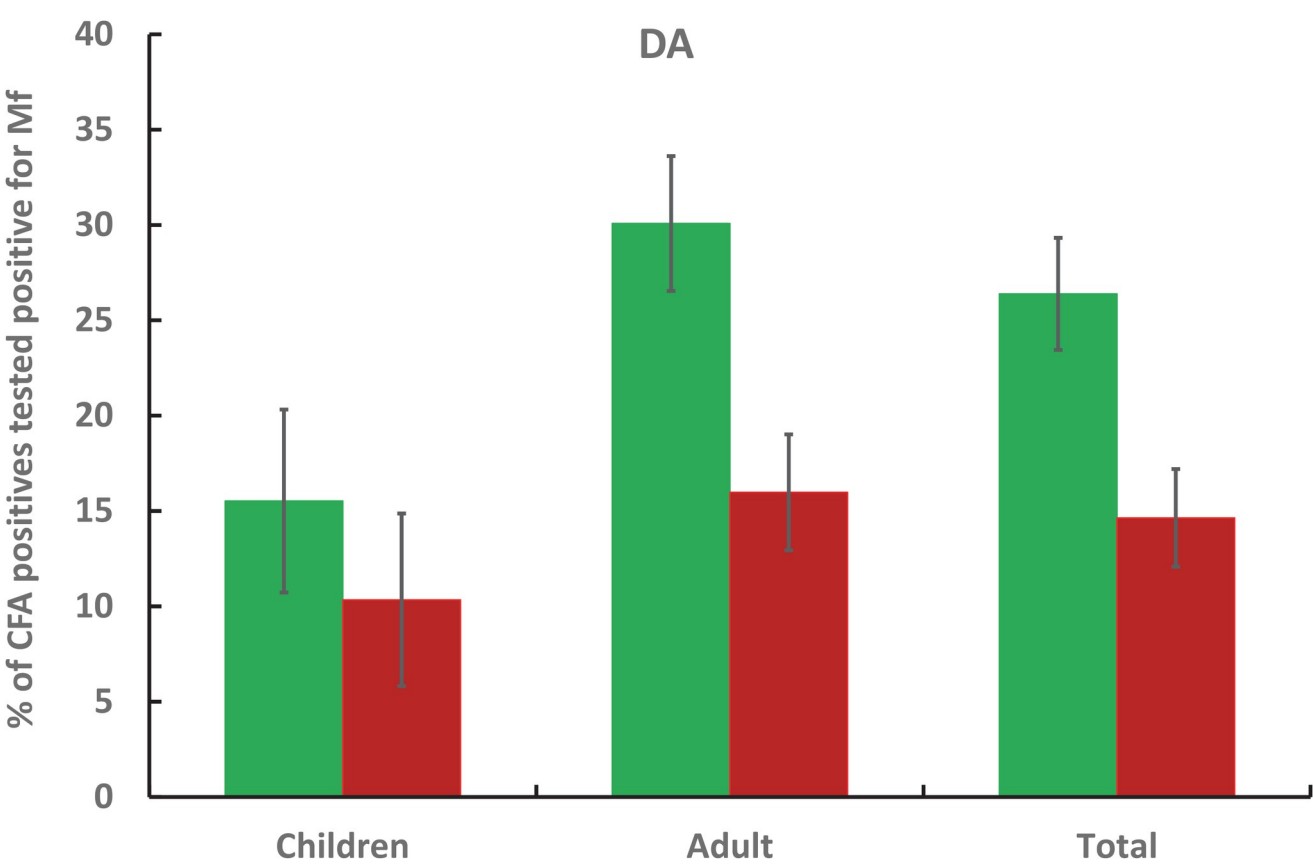

**Fig 8. Percentage of CFA positives tested positive for Mf prior to and 1-year post treatment with DA in children, adult and for all ages.**

certain proportion of Grade 1 AE from progressing to Grade 2. It is reassuring that most AEs were mild and transient. Rare localized reactions such as soft tissue swelling (in the scrotum or lymph nodes) and reversible limb edema reported earlier from this study is believed to be related to the death of adult filarial worms. Those AE also resolved with symptomatic treatment over time.

Mf carriers had a higher frequency of AE after IDA (40.5%) and after DA (20.3%) than Mf negatives. The frequency of AE after IDA was lower in microfilaremic individuals compared to that reported in clinical trials performed in Cote d'Ivoire (47%) and Papua New Guinea (83% and 59%)[5,6]. The geometric mean count of Mf in our study (IDA vs DA: 174.8 vs 141.5 per ml of blood, estimated based on Mf counted in the 60 μl blood smear) was comparable with that in Cote d'Ivoire study that compared IDA to IA (198 Mf/ml vs 190 Mf/ml) but lower than those reported from the IDA safety study in Haiti (IDA vs DA: 305 vs 223 per ml of blood) or from the IDA clinical trials in PNG (IDA vs DA 699 vs 744 Mf per ml blood) [6]; pilot study: (IDA vs DA:1558; vs 1857) [5]. The geometric mean Mf count in microfilaremic individuals who experienced AE in the India study was significantly higher than the mean Mf count in microfilaremic individuals who did not experience AE. Heavily infected individuals are more likely to experience AE after MDA, and the severity of adverse events has been reported to increase with increasing blood Mf counts. Fortunately, the severity of AE was not higher in persons with higher blood Mf counts in our study. The higher frequency of adverse events among microfilaremic individuals was likely related to death of Mf. However, this does not explain the higher frequency of AE in females, because infections were more common in

males in both treatment areas. It is likely that males tended to ignore or underreport mild AEs, and this has been as observed in other studies [23]. Adults had higher AE frequencies than children because of their higher infection prevalences. Significantly higher frequencies of AE among Mf-negative persons with CFA (9.2% after IDA and 7.4% after DA) compared to CFA-negatives (5% after IDA or DA) suggest that some persons with isolated CFA might have circulating Mf below the level detectable by thick blood smear examination.

Frequencies of AE reported from earlier clinical trials that were performed in India and elsewhere with ivermectin alone or with ivermectin in combination with either albendazole or diethylcarbamazine for treatment of Bancroftian or Brugian filariasis are not directly comparable with those of the present study because of differences in protocols for assessment of AE [24]. Similarly, wide variations in the frequency of AE in published reports of MDA programmes with DA in India (15.5–16.5%) and other countries may also be due to differences in monitoring protocols and differences in epidemiological settings. As reported by Weil et al. [13], the procedures used in the current study were based on the published recommendations for reporting AE in filariasis studies.

This study also compared the efficacy of IDA and DA for clearing filarial infections 12 months after treatment. IDA was much more effective than DA for clearing Mf, and this difference was also seen in prior clinical trials [6]. On the other hand, CFA clearance rates after IDA or DA in this study were not impressive (10.0–14.4%). CFA clearance after treatment was somewhat more frequent in those who had lower baseline FTS scores, particularly in children, and this is probably because of their worm loads at baseline. Although CFA clearance rates were higher in the Haiti IDA safety study (20.5–25.5%) than in this study, the Haiti study also found no difference in CFA clearance after IDA or DA [23]. In contrast to our results, clearance of CFA was higher among adults in Haiti. Neither DA nor IDA were effective for complete clearance of CFA in the randomized clinical trials performed in Papua New Guinea [5,6]. However, filarial antigen levels assessed by ELISA decreased more after IDA than after DA in one of those studies [5]. A recent study showed that while IDA treatment achieved sustained Mf clearance in more than 90% of Mf carriers, 75% of IDA recipients were still CFA positive five years after treatment [25]. These and other published results suggest that CFA may not be an appropriate indicator for post-MDA surveillance in areas where IDA is used for LF elimination. Therefore, different infection measures or modified surveillance strategies will be needed for MDA stopping decisions in areas where IDA is used.

Mf clearance rates 12-month post-treatment with either IDA (84%) or DA (62%) were lower in this study than those reported from the parallel study in Haiti (IDA vs DA: 94.4% vs 75.9) [23]. It is possible that this is due to the fact that DEC dosing was based on weight in Haiti and by age in our study. Compared to clinical trials in Papua New Guinea (IDA vs DA: 96% vs 32%) [6] and (100 vs 8.3%) [5], Mf clearance was lower after IDA and higher after DA in this study. However, Mf clearance rates in our study were higher than those reported from a clinical trial conducted in the Ivory Coast (71% after IDA; 26% after ivermectin plus albendazole) [26]. The significantly greater decrease in the percentage of CFA positive tested positive for Mf from baseline to post 1-year treatment after IDA than after DA in this study is consistent with the fact that IDA was more effective than DA for clearing Mf.

Pre-treatment Mf counts in microfilaremic individuals in our study were much lower compared to those in prior clinical trials or in the community IDA safety study in Haiti [23]. IDA and DA were almost equally effective in this study for clearing Mf when baseline Mf counts were low, while IDA was much more effective for clearing Mf in persons with higher baseline Mf counts. Thus, the added benefit of IDA will be most dramatic in areas where many people have high Mf counts. The relatively lower community Mf loads at baseline in this study may reflect many prior rounds of MDA with DA in Yadgir district. However, IDA was superior to

DA for clearing Mf in this setting despite extensive prior MDA, and IDA is likely to be useful for clearing other problem areas with residual LF in India with treatment histories that are similar to Yadgir's.

As a community-based, open label trial, the study had some limitations. One of the reasons for the lower Mf clearance in this study might be due to the difficulty in ensuring that all the participants swallowed all the tablets provided to them during night hours in a village setting. Our study showed that compliance rates (IDA: 73.5% Vs DA: 82.9% of eligible village residents) were at least comparable with available independent coverage survey results in Yadgir with two drugs (90.1% and 75.4% of the eligible population) after 8 and 15 rounds of MDA in 2011 and 2018) and higher than that reported after 11 rounds of MDA in 2015 (56.2%) [14–16]. Indeed, many residents of our study villages reported never having swallowed MDA drugs prior to this study. We believe that coverage can be improved with intensive IEC, community preparation and mopping up activities when IDA is introduced. Longer term follow-up of the CFA and Mf positives would have been helpful to confirm the sustained effect of IDA on adult worms, but routine MDA with DA was provided by health authorities in Yadgir district shortly after our 12-month follow-up evaluation.

Another potential challenge for this type of study is underreporting of AE that may have occurred in persons who were not available for post-MDA assessments. However, very high rates of follow-up assessment were achieved. Persons who were not present for AE assessments were contacted by mobile telephone to make sure that no significant AE were missed. Most people who were absent for AE assessments either had no symptoms or mild symptoms that did not interfere with their work or doing activities away from home.

Safety is a major concern in introducing a new drug regimen, especially when it is being considered for MDA. The current study showed that ivermectin at a dose of 200 μg/kg body weight was well tolerated when it was added to the standard double drug regimen of DEC and albendazole as an MDA regimen for LF elimination. IDA was much more effective than DA for clearing and suppressing Mf for at least one year, and the large size of the study lend credibility to the safety and efficacy results. We think that health officials should try to provide an effective and visible system for detecting and managing AE when IDA is first introduced in new areas to increase the confidence of the people in the new MDA regimen. IEC messaging should mention the added benefits of ivermectin against intestinal worms and ectoparasites such as lice and scabies.

Based on results from this study from India and parallel safety and efficacy results obtained in other countries, WHO now recommends IDA for LF elimination in specific situations (e.g., persistently high Mf or antigen prevalence after several rounds of routine MDA) and for LF elimination in newly identified endemic areas. IDA holds great promise as a "next generation MDA tool", but more data should be collected to document the benefits of IDA when it is used in true programme mode. The Government of India has recently provided MDA with IDA for millions of residents in five health districts, and plans are in place to roll it out in many other districts to help accelerate the final phase of India's LF elimination programme.

## Supporting information

**S1 Protocol. A community based study, to compare the safety, efficacy and acceptability of a triple drug regimen(Ivermectin, Diethylcarbamazine and Albendazole) with a two drug regimen(Diethylcarbamazine and Albendazole) for lymphatic filariasis elimination programme.**
(PDF)

**S1 Table. Participant follow-up rates for assessment of adverse events after treatment.**
(DOCX)

**S2 Table. Type of adverse events by severity and treatment regimens.**
(DOCX)

**S3 Table. Results of univariate and multivariate analysis of the association of CFA clearance with drug regimen, FTS score at baseline and demographic variables.**
(DOCX)

**S1 Fig. Frequency of the most commonly observed symptoms in Mf-positives among IDA and DA treated subjects.**
(TIF)

**S2 Fig. Frequency of the most commonly observed symptoms in Mf-negatives among IDA and DA treated subjects.**
(TIF)

## Acknowledgments

We thank Dr. Soumya Swaminathan, former Secretary and Director-General, Indian Council of Medical Research, Department of Health Research, Ministry of Health and Family Welfare, Govt. of India, for supporting this initiative and Dr Rashmi Arora, former Head and Scientist-G, Division of Epidemiology & Communicable Diseases ICMR New Delhi, for her support. We are thankful to Drs. P.K. Sen, former Director and Nupur Roy, Joint Director of National Vector Borne Disease Control Program for their support during the entire period of our study. We thank Dr. B.G. Prakash Kumar, Deputy Director, National Vector Borne Disease Control Program, Karnataka and Dr Divakar Malge, District Health Officer, Yadgiri for local support at the study site.

We thank Dr Yeramalli Subramanian of CliniOps, USA and Dr. Anand Eswariah, Vinayak Desai and Santosh Wale of Syngene International Ltd, Bengaluru, India, Dr. Ramakrishna Rao and Ms. Katiuscia O'Brian from Washington University in St. Louis for technical support for work with the electronic data capture system, for data monitoring and for training of project staff. Mr. Andrew Majewski from the Taskforce for Global Health in Decatur, Georgia, USA provided key administrative support. We also thank Dr. R L J De Britto, Dr. B. Nandha and Dr. A. Krishna Kumari of ICMR-Vector Control Research Centre for their support in community mobilization and morbidity management of chronic cases and Mr.Y. Srinivasamurthy of ICMR-Vector Control Research Centre for technical assistance. We thank the medical officers, nurses, technicians, field workers and data entry operators of this project for field work and data collection. We thank the medical officers and other PHC staff as well as ASHA workers and Anganwadi staff from study villages for assistance in field work and management of adverse events. We extend our thanks to community leaders (Mr. Sharanappa- Hatikkuni village, Mr. Parvath Reddy-Kandkur village, Mr. Sabayya- Gunjanur village, Mr. Gopal Reddy-Anpur village and Mr. Santosh-Nazrapur village) and volunteers for their involvement in IEC activities and motivation of villagers. Thanks, are also due to Julie Jacobson from the Bill & Melinda Gates Foundation USA, Dr. Bhupendra Tripathy, and Mr. Harish Iyer from the India Office of the Bill & Melinda Gates Foundation for facilitating this study.

Thanks are also extended to Prof. N. K. Ganguly, former Director General of ICMR as the Chairperson of National Filariasis Steering Committee and its members for providing valuable suggestion while reviewing the work, and to Dr. S. Sandhiya, Assistant Professor, Department of Clinical Pharmacology, Jawaharlal Institute of Postgraduate Medical Education and

Research, Puducherry, who served as the Medical Monitor for this study. The members of the Data Safety Monitoring Board (DSMB) are also acknowledged for their valuable suggestions. We are indebted to the late Dr. V. Kumarasamy, former Director of ICMR- National Institute for Research on Tuberculosis, Chennai, who had a leading role in the early planning stages of this study.

## Author Contributions

**Conceptualization:** Purushothaman Jambulingam, Kaliannagounder Krishnamoorthy, Swaminathan Subramanian, Gary J. Weil.

**Data curation:** Swaminathan Subramanian, Adinarayanan Srividya, Hari Kishan K. Raju.

**Formal analysis:** Swaminathan Subramanian, Adinarayanan Srividya.

**Funding acquisition:** Purushothaman Jambulingam, Gary J. Weil.

**Investigation:** Purushothaman Jambulingam, Vijesh Sreedhar Kuttiatt, Kaliannagounder Krishnamoorthy, Swaminathan Subramanian, Adinarayanan Srividya, Manju Rahi, Roopali K. Somani.

**Methodology:** Purushothaman Jambulingam, Vijesh Sreedhar Kuttiatt, Kaliannagounder Krishnamoorthy, Swaminathan Subramanian, Gary J. Weil.

**Project administration:** Purushothaman Jambulingam, Vijesh Sreedhar Kuttiatt, Kaliannagounder Krishnamoorthy, Swaminathan Subramanian, Manju Rahi.

**Resources:** Purushothaman Jambulingam, Manju Rahi, Gary J. Weil.

**Supervision:** Purushothaman Jambulingam, Vijesh Sreedhar Kuttiatt, Kaliannagounder Krishnamoorthy, Swaminathan Subramanian, Adinarayanan Srividya.

**Validation:** Swaminathan Subramanian, Adinarayanan Srividya.

**Visualization:** Hari Kishan K. Raju.

**Writing – original draft:** Purushothaman Jambulingam, Vijesh Sreedhar Kuttiatt, Kaliannagounder Krishnamoorthy, Swaminathan Subramanian.

**Writing – review & editing:** Purushothaman Jambulingam, Vijesh Sreedhar Kuttiatt, Kaliannagounder Krishnamoorthy, Swaminathan Subramanian, Adinarayanan Srividya, Hari Kishan K. Raju, Manju Rahi, Roopali K. Somani, Mallanna K. Suryaprakash, Gangeshwar P. Dwivedi, Gary J. Weil.

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
