## [Decision Letter · Decision Letter 0]

19 Aug 2020

Dear Dr. Swaminathan,

Thank you very much for submitting your manuscript "An open label, block randomized, community study of the safety and efficacy of co-administered ivermectin, diethylcarbamazine plus albendazole vs. diethylcarbamazine plus albendazole for lymphatic filariasis in India" for consideration at PLOS Neglected Tropical Diseases. As with all papers reviewed by the journal, your manuscript was reviewed by members of the editorial board and by several independent reviewers. In light of the reviews (below this email), we would like to invite the resubmission of a significantly-revised version that takes into account the reviewers' comments. 

We cannot make any decision about publication until we have seen the revised manuscript and your response to the reviewers' comments. Your revised manuscript is also likely to be sent to reviewers for further evaluation.

Sincerely,

Subash Babu

Guest Editor

Sara Lustigman

Deputy Editor

Reviewer's Responses to Questions

**Key Review Criteria Required for Acceptance?**

**Methods**

-Are the objectives of the study clearly articulated with a clear testable hypothesis stated?

-Is the study design appropriate to address the stated objectives?

-Is the population clearly described and appropriate for the hypothesis being tested?

-Is the sample size sufficient to ensure adequate power to address the hypothesis being tested?

-Were correct statistical analysis used to support conclusions?

-Are there concerns about ethical or regulatory requirements being met?

Reviewer #1: The objectives of the study are very clearly articulated, and the study design is appropriate for these objectives. The study population is clearly described, and appropriated. The sample size is adequately powers, and the statistical analyses used are appropriate. I have no concerns about meeting ethical or regulatory requirements.

Reviewer #2: Yes to above. Please address the following points:

• Line 146: How were the villages split into the two blocks? Why 4 villages in one block and 2 in the other?

• Lines 170-171: How were these assumptions determined?

• Line 175: For readers who are not familiar with India, it would be helpful to describe what anganwadi means.

• FTS: Was testing done during household visits, or in a field lab?

• Line 296: This sentence needs more explanation: “The EDC system is 21 CFR Part 11 compliant”

• Age groups: The methods described analysis by two age groups - child (<18) or adults. Was it possible to conduct analyses by smaller age groups? E.g. Paper by Dubray et al used age 5-9, 10-17, and 18+. https://journals.plos.org/plosntds/article?id=10.1371/journal.pntd.0008298

Reviewer #3: -Are the objectives of the study clearly articulated with a clear testable hypothesis stated? Yes

-Is the study design appropriate to address the stated objectives? Yes

-Is the population clearly described and appropriate for the hypothesis being tested? Yes

-Is the sample size sufficient to ensure adequate power to address the hypothesis being tested? Yes

-Were correct statistical analysis used to support conclusions? Yes

-Are there concerns about ethical or regulatory requirements being met? No

**Results**

-Does the analysis presented match the analysis plan?

-Are the results clearly and completely presented?

-Are the figures (Tables, Images) of sufficient quality for clarity?

Reviewer #1: The results are very clearly and comprehensively presented. The analyses presented in the Results section match the plan described in the Methods section. The tables and images are generally of sufficient clarity, except for Figure 3 (Forest plot), in which the font size is too small for easy reading.

Reviewer #2: Generally yes, please address the following points:

• How many households were visited in total? Average/range of number of participants per household?

• Table 2: Results by age groups should be included

• Table 3: Results are shown by three age groups here (different to what was described in Methods and the text lines 372). The first two age groups overlap (6-8 vs 5-17), which is a bit confusing. Why were results for 6-8 year olds reported separately in this table but not done anywhere else?

• Lines 379-380: Please point the reader to the results, and briefly describe them here. i.e. what % were Grade 1 and 2 for each arm.

• Table 5 is described in the text before Table 4. 

• Lines 384-387: Briefly describe key results here, including numbers.

• Figures 4: please add 95% CI error bars.

• Figure S1: please add 95% CI error bars. If the main focus of this study is to compare DA with IDA, it would be better to have IDA vs DA for Mf positives in one panel, and IDA vs DA for Mf negatives in another panel.

• Line 424-426: Some numbers here for the key results would be helpful. Was the difference in fever significant between IDA and DA?

• Table S2: There is an error in the partitioning of the first line of the table.

• Lines 438-442: Some numbers here would be helpful.

• Line 444: How many developed these reactions? How long did they last for? Did any last for longer than the 7 day follow up period?

• Figure 6: How does this compare to the age/sex distribution of those who were still Mf+ after 1 year?

• Line 488: Some may not consider this a “small” difference.

• Fig 7: I don’t think this Figure is necessary – results were the same for all groups and can just be described in text.

Reviewer #3: -Does the analysis presented match the analysis plan? Yes

-Are the results clearly and completely presented? Yes

-Are the figures (Tables, Images) of sufficient quality for clarity? Yes

**Conclusions**

-Are the conclusions supported by the data presented?

-Are the limitations of analysis clearly described?

-Do the authors discuss how these data can be helpful to advance our understanding of the topic under study?

-Is public health relevance addressed?

Reviewer #1: The conclusions are well supported by the results that are presented, and the limitations are discussed comprehensively. The public health relevance is very clearly explained by the the authors discussion of how the data presented in this paper leads us to a greater understanding of the adverse events and efficacy of using IDA as a tool for the elimination of LF.

Reviewer #2: Generally yes. Please address the following points:

• The discussion is very long and some sections are repetitive. Could be made more concise.

• Line 526: What were the rates at the other sites?

• Line 558: reference needed.

• Line 576: reference needed.

• Lines 587-588: Belongs in results

• Lines 606-609: Although true, this is not really relevant to the current study, and could be removed.

• Although IDA was more effective than DA for clearing Mf at one year, 16% of those who received IDA were still Mf positive (and therefore infectious) after one year. Was IDA expected to be more effective than this? 

• Similarly, DA was only 61.8% effective at clearing Mf at one year – based on many years of experience with DA, was this result expected?

• Considering that neither DA or IDA reduced CFA significantly after one year, some comments about the use of CFA in post-MDA surveillance would be helpful.

Reviewer #3: -Are the conclusions supported by the data presented? Yes

-Are the limitations of analysis clearly described? Yes

-Do the authors discuss how these data can be helpful to advance our understanding of the topic under study? Yes

-Is public health relevance addressed? Yes

**Editorial and Data Presentation Modifications?**

Reviewer #1: A few suggestions are made for improvement of this manuscript:

1. In the Methods section, subsection on Study Location (p 7), please provide some data on MDA coverage in the relevant LF MDA Implementation Unit, in the round(s) of MDA prior to initiation of the study. This would provide the relevant context for the statement made in the Discussion that "The study was carried out in an area where compliance for routine MDA for elimination of LF was very low" (lines 623 - 624). 

2. In the Methods section, subsection on Testing for CFA and Mf, it is stated that “CFA-positive participants were screened for Mf after 9.00 PM by night blood smear examination” (line 220). This implies that only CFA positive participants were screened for Mf by NBS. However, data provided in Table 3 suggests that almost all who were tested for CFA (total of 4732) were also screened for Mf (4706). This discrepancy should be clarified. 

3. In the Methods section, subsection on Clinical monitoring and assessment of adverse events, it is stated that "Adverse events were recorded as mild (grade 1), moderate (grade 2) and severe (grade 3)" (lines 256 – 257). However, Table 5 includes an additional grade (Grade 4 AE). Please clarify.

4. The CONSORT diagram in Figure 2 indicates that non-compliance was much higher in the IDA arm (n=1989) than in the DA arm (n=860). As a proportion of the eligible population in the two arms, this is approximately 30% vs 17%. Some comment on the reasons for this difference in the discussion might be useful for guiding future strategies for the adoption of IDA for LF elimination in India.

Reviewer #2: Please see above comments regarding figures.

Reviewer #3: (No Response)

**Summary and General Comments**

Reviewer #1: This is a very well-written paper that provides detailed information from one study site in a multi-centre study that assessed IDA as a tool in the global programme for the elimination of lymphatic filariasis. The information provided here on adverse events associated with use of IDA and its efficacy in clearance of Mf and CFA 12 months after treatment, is a valuable addition to the evidence base for adoption of this important new tool.

Reviewer #2: This is an important study about adverse reactions and efficacy of 2-drug vs 3-drug MDA in India. The study is well designed, and the paper is well written. The results will contribute significantly to LF elimination efforts. The paper is suitable for publication in PLOS NTDs after the following points have been addressed.

Abstract

• Line 35: Not all AEs were mild, please include %s.

• Line 37: was this difference significant?

Introduction:

• Lines 72 and 76: different numbers of LF endemic countries (72 vs 73)?

• Line 95: Why were some districts “hard core”? Poor coverage, other reasons?

• Line 103: Would be helpful to mention where this larger trial was done, and on how many people.

• Line 112-114: As mentioned later in the manuscript, WHO started recommending large-scale use of IDA from 2017, and countries started implementing this in 2018.

Reviewer #3: The manuscript is one of the major studies that the authors have carried out using the triple drug therapy (Ivermectin/DEC/Albendazole), in continuation of previous studies in other places.

The manuscript is excessively long (but understandably) and might benefit from removing redundant text, that is also present in the detailed supplementary text. No major critiques. This is a massive effort and commendable for the efforts put in. The numbers have been a bit confusing though. Although it might be difficult to compile, a single excel sheet with all of the data could be more useful. 

Comments:

1. The methods section could be greatly trimmed, as most of it is also included in the supplementary text. 

2. Ln 354. 70.1 % received treatment with IDA and 74.9 % received DA. Going to the consort diagram, we see that 4758 of 6475 individuals received IDA (prior to AE assessment). This amounts to 73.48% of eligible individuals. Likewise, 4160 of 5020 individuals received DA (prior to AE assessment), that amounts to 82.86%. Even when the numbers of the ‘Full analysis set included’ are taken, the percentages are 72.2 % and 82.19 %. 

3. Further, while the CONSORT diagram excludes 2 individuals each in the IDA and DA arms, the corresponding histogram (Figure 4) with all the AE’s does not exclude them.

4. Again, in the same CONSORT diagram, doing the math for the excluded individuals (in the box to the top right), 6785 – 38 = 6741 for IDA; so not sure where the 6475 comes from. So assuming 6475 is the right number, excluding non-compliant in IDA (n=1989), we should get 4486 (6475 – 1989 = 4486) instead of 4758. The discrepancies need to be sorted out.

5. Table 2. 326 (6.9%) in IDA and 265 (6.4%) in DA arm were MF positive before treatment. However, the CONSORT figure (Figure 5), has them as 324 and 264.

6. Similarly, the number of people screened for MF were 4706 (IDA) and 4119 (DA), for a total of 8825 individuals. However, the CONSORT figure (Figure 5) states that the number of people tested for MF prior to treatment were 8805. 

7. Figure 4. Footnotes state “The overall AE rates or severity of AE differ by treatment regimens (p<0.01) for all comparisons”. If this is true, then all the comparisons made should be significant across the study?

8. Not sure if Figures 6 & 7 adds anything to what is already described in the text and tables and can be removed.

9. Tables 5 & 6 – The last two columns can be deleted. A simple sentence that no AE’s beyond grade 3 were observed.

10. Most tables have ‘excluded for lack of data for either AE’s, MF or CFA’. Maybe a column can be added to the table(s) to include the number that were excluded, so that the numbers across all the tables/figures match up and easy to interpret.

11. One of the objectives of the study was “To assess and compare the prevalence of antibody with that of Ag and MF” (Supplementary information). However, I do not seem to find any findings related to this objective.

12. Will the data be available for the community for any further rmetanalyses?

PLOS authors have the option to publish the peer review history of their article (what does this mean?). If published, this will include your full peer review and any attached files.

Reviewer #1: Yes: Nilanthi Renuka de Silva

Reviewer #2: No

Reviewer #3: No
---

## [Decision Letter · Decision Letter 1]

9 Nov 2020

Dear Dr. Swaminathan,

Thank you very much for submitting your manuscript "An open label, block randomized, community study of the safety and efficacy of co-administered ivermectin, diethylcarbamazine plus albendazole vs. diethylcarbamazine plus albendazole for lymphatic filariasis in India" for consideration at PLOS Neglected Tropical Diseases. As with all papers reviewed by the journal, your manuscript was reviewed by members of the editorial board and by several independent reviewers. The reviewers appreciated the attention to an important topic. Based on the reviews, we are likely to accept this manuscript for publication, providing that you modify the manuscript according to the review recommendations. 

Sincerely,

Subash Babu

Guest Editor

Sara Lustigman

Deputy Editor

Reviewer's Responses to Questions

**Key Review Criteria Required for Acceptance?**

**Methods**

-Are the objectives of the study clearly articulated with a clear testable hypothesis stated?

-Is the study design appropriate to address the stated objectives?

-Is the population clearly described and appropriate for the hypothesis being tested?

-Is the sample size sufficient to ensure adequate power to address the hypothesis being tested?

-Were correct statistical analysis used to support conclusions?

-Are there concerns about ethical or regulatory requirements being met?

Reviewer #1: Yes to all of the above.

Reviewer #2: 1. Thanks for providing additional details about the sampling strategy:

• Lines 158+: “However, Mf surveys carried out in 2016 (prior to this study), as a part of the programme monitoring and evaluation, showed that the Mf-prevalence was above 1% in in all four sentinel and four spot check sites in the district. In addition to these eight sentinel and spot-check sites, Mf surveys were carried out by our team in five other high-risk villages/wards (sites) prior to inititation of the study. Mf prevalence in these thirteen sites ranged from 1.1 to 7.1% (total N surveyed, 7,589).”

and

• Line 169+: “The results of Mf-survey data collected prior to the study were used to identify two blocks of 4 and 2 villages that were comparable with respect to Mf-prevalence and population size (Figure 1).”

However, it is still unclear how the survey villages were selected. The study design and village selection need to be made more clear so that any biases can be appropriately assessed, particularly when the study has found some unexpected results.

• It’s still not clear why 4+2 villages were chosen (e.g. instead of 3+3 or 5+1). It seems odd for 4+2 to have been a pre-determined plan. Or was this decided once the results of the Mf survey (before the start of this study) were available, and the villages already chosen, then split into 4+2 to give approximately equal numbers in each arm? 

• How were the additional five high-risk villages selected? Were they purposefully selected because of known previous high prevalence? 

• Is it possible to provide more specific details about the Mf prevalence in the sentinel and spot check sites? 

2. Thanks for providing additional information about households in the response, but I can’t see this information in the manuscript: “1652 (77.1%) of the 2149 households in the IDA arm and 1260 (86.4%) of 1459 households in the DA arm were visited. The average (range) no. of participants per household were 4.6 (1-21) and 4.7 (1-20) in the respective arms.” 

Analyses should have taken into account the number of participants per household, particularly because clustering of CFA-positive and Mf-positive individuals within households have been identified in LF studies elsewhere. Considering that there was a large range in the number of household participants (1-21 for IDA and 1-20 for DA), household-level clustering could have significantly affected prevalence estimates and confidence intervals for CFA and Mf. Clustering should therefore be assessed (e.g. using intra-cluster correlation - ICC), and prevalence estimates for CFA and Mf adjusted for household clustering if necessary. I note that the mixed effect logistic regression model used village (cluster) as a random effect, but did not consider household. Adverse events might also cluster within households, especially subjective symptoms.

Reviewer #3: (No Response)

**Results**

-Does the analysis presented match the analysis plan?

-Are the results clearly and completely presented?

-Are the figures (Tables, Images) of sufficient quality for clarity?

Reviewer #1: Yes to all of the above.

However, as in Table 3, Table 2 should also indicate that the total numbers given as screened for microfilaraemia include those who were CFA negative, and therefore not subjected to night blood testing

Moreover, there is some disparity in the numbers presented in Tables 2 and 3 and in Figure 5. Tables 2 and 3 indicate that a total of 8887 persons were screened for CFA (4155 DA + 4732 IDA) whereas the CONSORT diagram in Figure 5 states that 8914 persons were tested for CFA prior to treatment. 

Similarly, Table 2 indicates that a total of 8825 persons were screened for mf (4119 DA + 4706 IDA), but Figure 5 indicates that 8805 persons were screened for mf prior to treatment. These discrepancies should be rectified.

Reviewer #2: Yes, but please refer to my comments above under Methods. Some of the prevalence estimates might need to be amended.

Reviewer #3: (No Response)

**Conclusions**

-Are the conclusions supported by the data presented?

-Are the limitations of analysis clearly described?

-Do the authors discuss how these data can be helpful to advance our understanding of the topic under study?

-Is public health relevance addressed?

Reviewer #1: Yes to all of the above

Reviewer #2: Yes

Reviewer #3: (No Response)

**Editorial and Data Presentation Modifications?**

Reviewer #1: A few typographical errors need correction:

Page 6 of the pdf document, line 4: 'Krishnamoorthy' written as 'Krishnamoorhty'

Page 12, line 141: 'study' written as 'srudy'

Page 20, line 345: p is missing in 'positive'

Page 25, line 414: missing bracket

Page 33, line 567: 'comparable' written as 'comparbable'

Figure 6, x-axis label: 'prior' written as 'priror'

Reviewer #2: Typo in line 389 – should be CFA-positive.

Reviewer #3: (No Response)

**Summary and General Comments**

Reviewer #1: This revised manuscript addresses most of my concerns, except for the matter described above

Reviewer #2: Many thanks to the authors for providing comprehensive responses to my questions. Again, this is an important study and will contribute significant knowledge for the Global Programme to Eliminate LF, and I congratulate the authors on the excellent work. I have two remaining questions regarding methods (as described above) - I believe that it is important to address these questions before acceptance for publication.

Reviewer #3: I do not have any major comments. The comments raised earlier appears to have been satisfactorily addressed.

PLOS authors have the option to publish the peer review history of their article (what does this mean?). If published, this will include your full peer review and any attached files.

Reviewer #1: No

Reviewer #2: No

Reviewer #3: No
---

## [Editor Report · Decision Letter 2]

12 Dec 2020

Dear Dr. Swaminathan,

We are pleased to inform you that your manuscript 'An open label, block randomized, community study of the safety and efficacy of co-administered ivermectin, diethylcarbamazine plus albendazole vs. diethylcarbamazine plus albendazole for lymphatic filariasis in India' has been provisionally accepted for publication in PLOS Neglected Tropical Diseases.

Best regards,

Subash Babu

Guest Editor

Sara Lustigman

Deputy Editor

---

## [Editor Report · Acceptance letter]

8 Feb 2021

Dear Dr. Swaminathan,

We are delighted to inform you that your manuscript, "An open label, block randomized, community study of the safety and efficacy of co-administered ivermectin, diethylcarbamazine plus albendazole vs. diethylcarbamazine plus albendazole for lymphatic filariasis in India," has been formally accepted for publication in PLOS Neglected Tropical Diseases.

Best regards,

Shaden Kamhawi

co-Editor-in-Chief

Paul Brindley

co-Editor-in-Chief
